# BUILDING LEARNING CONTEXT FOR AUTONOMOUS AGENTS THROUGH GENERATIVE OPTIMIZATION

## ABSTRACT

Building intelligent agents that learn involves designing systems that can evolve their behavior based on experiences. While early approaches to large language models (LLMs) agent learning relied mostly on structured memory and in-context learning, they often led to behavioral instability, poor interpretability, and difficulty in control. Recent success in generative optimization, where an LLM is used as an optimizer, has shown the possibility of creating autonomous software agents. By separating behavior logic (workflow) and how that logic is updated (optimizer), the agent designer can exhibit more control over the agent. In this work, we show the surprising fact that the agent learning problem is *under-specified* with the generative optimization framework. If we want an agent to learn the right behavior, we must set up the right context that will induce such behavior. We investigate three types of software engineering problems that span data science, computer security, game playing, and question answering. We show that the original generative optimization framework can only learn robustly under one of the three settings. To address the issue, we propose to construct a *meta*-graph through templates to introduce the right learning context to an LLM optimizer. With this addition, we demonstrate that defining the right learning context enables agents to discover behaviors aligned with the designer's objectives. In particular, we show the first known result of using generative optimizers to learn executable programs that play Atari games, where the resulting agents achieve performance comparable to deep reinforcement learning while requiring 50%-90% less training time.

## 1 INTRODUCTION

Artificial intelligent *agents* — computer programs that "operate autonomously, perceive their environment, persist over time, adapt to changes, create and pursue goals" (Russell & Norvig, 2016) — have regained significant attention recently, due to the maturation of large language models (LLMs) (Achiam et al., 2023). The ease of accessing LLMs gave rise to new programming paradigms, such as language model programs (Khattab et al., 2024) and multi-agent orchestration frameworks (Wu et al., 2024), all of which leverage calls to LLMs to handle a wide range of tasks in human society, from computer use (Fourney et al., 2024), and software engineering (Jimenez et al., 2024), to scientific discovery (Yang et al., 2024b).

Despite these advances, building agents still requires a substantial amount of human engineering. Often agent developers need to design complex decision rules to orchestrate an agent's behaviors, build pipelines to parse information from the environment into the agent's percepts, and engineer prompts to control LLMs. However, results in competitive programming (El-Kishky et al., 2025) have shown that it is paramount to find a general-purpose method that can scale with compute, rather than engineering domain-specific solutions (Sutton, 2019). These human-engineered components should be automatically *learned* through agent's experiences, enabling agents that can program themselves through learning.

Traditional techniques such as reinforcement learning (RL) (Sutton et al., 1998) are general-purpose algorithms that theoretically can be applied to agent learning. However, since state-of-the-art agents are made of large language models (LLMs), gradient updates must be performed on LLM weights. Initial explorations showed promising results (Bai et al., 2025; Guo et al., 2025), although it is unclear how well the learned LLM weights generalize to out-of-domain tasks, and how many tasks are needed

**Figure 1: A Diagram for An LLM-based System.** ■ represents a repeated workflow execution graph (we denote as *workflow graph*). We use ■ and ■ to represent trainable parameter nodes (string, code, etc.). The optimizer ■ updates parameter nodes during learning. We show that as agent designer, we can choose to optimize the parameters under different learning contexts (interactive, batch, and episodic). We show how to leverage repeatable *workflow graphs* (■) can be concatenated to construct an *agent learning graph* (■).

to reach a *generalist* agent, as multi-task learning has always been a challenge in RL (Kirkpatrick et al., 2017; Taiga et al., 2023).

Recently, there is an emerging end-to-end perspective that instead treats a given software program as a computational graph and optimizes its parameters (i.e., the human-made decisions discussed above) via a generalized form of "back-propagation gradient descent" (Cheng et al., 2024b; Yuksekgonul et al., 2025; Wang et al., 2024b). By using generative models (like LLMs) as optimizers (Pryzant et al., 2023; Nie et al., 2023; Yang et al., 2024a), this perspective has led to frameworks that can automatically tune parameters and generate new code in non-differentiable programs, which achieved promising results in producing distributed systems programs (Wei et al., 2024).

This approach offers several advantages. It enables an agent to adapt to solve tasks through automatic optimization, directly using rich feedback (such as compilation errors, system reports, and end-user feedback), a process that traditionally requires manual trial and error. Moreover, by learning, agents can discover solutions that exceed what human experts can design. We have seen this trend with deep learning (Silver et al., 2016; Brown & Sandholm, 2019; Mirhoseini et al., 2020; Bellemare et al., 2020). This perspective turns agent learning into a closed-loop optimization problem by calling LLMs over multiple iterations to refine the agent. This is in contrast to the predominant use of LLM-as-agents when the agents change their behaviors based on each query and hope to produce the correct sequence of actions in one-shot.

Prior works that incorporate environment feedback to change an agent's behavior have met with mixed success. While there are very successful applications (Cheng et al., 2024b), limitations have also been observed: the optimization process can be unstable (Huang et al., 2024) and the self-improvement phenomenon only persists for a few rounds (Shinn et al., 2023; Madaan et al., 2023). We argue that this issue arises in part because the problem of agent learning is *under-specified* with generative optimization. An agent needs to learn solutions that can generalize different contexts, while generative optimization defines an optimization problem under a single problem context.

In this paper, we analyze sources of this under-specification issue and propose constructive remedies. We show that agent learning should not be specified only as an execution (workflow) graph of its own internal operations, but a *meta*-graph on a stream of experiences to capture the learning context. We introduce operators ($\oplus$ and $\Rightarrow$) to insert workflow graphs into a template to construct an agent learning graph, which correctly specifies the agent's learning objective and enables generative optimization to learn parameters effective for the agent designer's goal. In addition, we discuss how to structure the agent's internal workflow to improve optimization results (similar to how architectural choices in neural networks facilitate better learning outcomes). We note that this factor has been overlooked in previous attempts to design self-improvement loops (Chen et al., 2023; Huang et al., 2024; Snell et al., 2024). Finally, we discuss a few choices of enhancing and amplifying feedback for different stages of the learning process, analogous to reward shaping.

We show that these insights allow us to apply generative optimization to solve a wide range of tasks. Automatic software engineering, such as creating an agent to write machine learning programs, can be seen as an interactive learning task. We show that on the MLAgentBench (Huang et al., 2023), we can learn an agent that can output high-quality models that surpassed 86.6% submissions on Kaggle leaderboard than the baseline agent, which only surpassed 70.8% submissions. We can also optimize an LLM based workflow to improve its performance by as much as 14.5% on GSM8K and 65.1% on BBEH. Finally, we show that we can learn a static Python program that can play Atari games, nearly

matching the performance of Deep RL baselines but with 50%-90% less compute time. All of these show the versatility of this paradigm and the power of the automatic agent learning process.

## 2 BACKGROUND AND RELATED WORK

**History of Learning Agents**   The term AI agent has a long history (Genesereth & Nilsson, 1987). In this paper, we follow Russell & Norvig (2016) and define a learning agent as a program that can sense percepts, take actions, and adapt with experiences in a digital/physical environment. For an agent to learn, it implies that the software has components that are modifiable and can influence its behaviors; these components are called *parameters*. For example, a tabular agent has a lookup table as its parameter, and a learning algorithm such as policy iteration or value iteration would be suitable (Bertsekas, 1987). A deep RL agent has a neural network as its parameter, and learn from rewards using algorithms like proximal policy optimization (PPO) (Schulman et al., 2017).

**Adaptive Workflow**   Increasingly, intelligent systems are being built with LLMs. For the state-of-the-art LLM systems (Wang et al., 2024a;c; Fourney et al., 2024), their parameters can be model weights, or more generally, system prompts, code that pre-processes input, and code that modifies the returned results from the LLM. Besides the obvious approach of fine-tuning the LLM's weights (Scheurer et al., 2023), there isn't a dominant approach on how to change the system's behavior on the fly. Some inference-time learning methods have been introduced, with the prevailing strategy utilizing databases, referred to as "memories" in RAG (Lewis et al., 2020). Recently, a new perspective of building intelligent agent emerges, which leverages an LLM's ability to write coherent programs to accomplish a purpose (Cheng et al., 2024b; Zhang et al., 2024). This view separates an LLM agent into two parts: the workflow that represents the behavioral logic of the agent, and an optimizer that updates such behavioral logic.

**Generative Optimization**   Generative optimization algorithms have been proposed to update an LLM workflow. They typically use a generative model (like an LLM) as part of its optimizer to analyze problems and propose updates. A generative optimizer takes as inputs (1) a problem context, (2) parameters, (3) a computational graph involving the parameters, (4) a feedback signal, and outputs a parameter value. Several generative optimizer implementations have been proposed, such as DSPy (Khattab et al., 2024), OptoPrime (Cheng et al., 2024b), TextGrad (Yuksekgonul et al., 2025), and GASO (Wang et al., 2024b). They differ in how they represent and reason about the graph and kinds of feedback they can process. For instance, optimizers in DSPy work with scalar feedback, while OptoPrime/TextGrad/GASO uses any feedback that an LLM can interpret. OptoPrime formats the entire graph into a single LLM prompt, while TextGrad/GASO processes the graph iteratively.

**The Framework of OPTO**   Recently OPTO (Optimization with Trace Oracle) (Cheng et al., 2024b) was proposed as a unified math setup for describing iterative generative optimization problems. An OPTO problem (a generalization of numerical optimization) is described by a tuple $(\Theta, \omega, \mathcal{T})$, where $\Theta$ is the parameter space, $\omega$ is the problem context and $\mathcal{T}$ is a Trace Oracle. For a parameter $\theta \in \Theta$, the Trace Oracle $\mathcal{T}$ returns a tuple $(f, g)$ where $g$ is a computational graph involving $\theta$ and $f$ is a feedback signal provided to exactly one node of $g$ (the output node). An autonomous agent that learns through experience in this setup corresponds to a workflow design and an optimizer that can update the workflow. We emphasize that a workflow itself is not an autonomous agent, but a workflow combined with an optimizer that can rewrite its own behavior according to feedback is an agent.

## 3 BUILDING LEARNING AGENTS WITH GENERATIVE OPTIMIZATION

In agent learning, we wish to optimize an agent's parameters in a stream of experience. Our approach takes inspiration from deep learning, which accomplishes machine learning via optimization on differentiable computational graphs. In deep learning, we specify a neural network architecture (a computational graph which is parameterized by tensors) and a numerical oracle (e.g., a loss function to minimize) to provide feedback at the output of the computational graph. Following the OPTO framework, our approach is built similarly with these two components but using generative optimization. The main differences are that here differentiability is not required and that the agent is not limited to learning from numerical feedback only. In the following, we show constructive

templates to define the parameterized computational graph and discuss design principles for the feedback oracle and the agent's computational graph.

We demonstrate how computational graphs can naturally describe different agent learning problems. We suppose that a workflow is given and is represented as a computational graph $W_\theta$, where $\theta$ denotes the parameters. Without loss of generality, we suppose the workflow takes a single input $x$ and returns a single output $y = W_\theta(x)$ ($x, y$ can be arrays for modeling multi-input-multi-output cases). We call $W_\theta$ the *workflow graph*. Recall that the workflow here means the full software program, which internally may be composed of multiple calls to LLMs and decision rules (in other words, with abuse of notation, the workflow here can represent also an entire multi-agent orchestration (Wu et al., 2024)). As a result, the workflow graph does not need to be static and it can vary with input or due to the internal randomness of the workflow.

In contrast to the workflow, we denote the full computational graph that will be presented to the generative optimizer (i.e., $g$ in OPTO) as $A_\theta$, which we call the *agent learning graph*. The agent learning graph is composed of the workflow graph and other nodes derived from the learning problem, such that the learning structure can be captured correctly. Lastly, a feedback oracle maps $(x, y)$ into feedback $f$, which can be numerics, texts, images, or structured objects (e.g. a dictionary). We assume the feedback is not adversarial and contains some information of the agent's performance. With these assumptions, an OPTO problem instance can be created where the problem context $\omega$ can be fixed to a string such as "Update the parameters to incorporate the feedback."

Now we discuss how to construct the agent graph for common agent learning problems using templates to build *agent learning graphs*, as shown in Fig. 1.

**Interactive Learning Template.** Here the agent learns on the fly as it interacts with the world (Shalev-Shwartz et al., 2012). At each time step, it sees an input $x$, outputs $y$, receives feedback $f$, and then updates its parameter $\theta$. These problems encompass online learning and bandit variants that are well-studied in the literature. The meta graph here simply shows the input $x$ is transformed by an operator (i.e., the agent) and then the feedback is provided to $y$. When the agent graph $A_\theta$ is inserted into this template, it yields the workflow graph $W_\theta$.

**Batch Learning Template.** Different from interactive learning, a batch-learning agent learns from a given dataset $D = \left[(x_i, z_i)\right]_{i=1}^N$ of size $N$, where $x_i$ and $z_i$ denote the input and the information to learn from for the $i$th data point (Hastie et al., 2009). For example, $z_i$ can be the desired agent output when seeing $x_i$ (supervised learning), or it can be a positive-negative pair (preference-based learning), etc. In batch learning, the agent learns from an oracle that takes $(x_i, y_i, z_i)$ as input (where $y_i = W_\theta(x_i)$) and provides feedback such as a loss. To handle such a batch problem with the iterative setup of OPTO, we appeal to the idea of online-to-batch conversion (Shalev-Shwartz et al., 2012) and mini-batching. As shown in Fig. 1, in each iteration of OPTO, we sample a minibatch and construct the graph for the sampled batch. We introduce a batchify operator $\oplus$ that concatenates different inputs. For a minibatch $\left[(x_i, z_i)\right]_{i=1}^B$ of size $B$, we first obtain $\left[(f_i, g_i)\right]_{i=1}^B$, where $g_i$ and $f_i$ result from input $x_i$. Suppose $o_i$ is the output node of $g_i$. Then we concatenate the outputs from $\{g_i\}_{i=1}^B$ to create a new node $\hat{o} = \oplus_{i=1}^B o_i$ and give the concatenated feedback $\oplus_{i=1}^B f_i$ to $\hat{o}$.

**Episodic Learning Template.** Here we adopt a broader definition of RL, which describes the agent learning in a sequential decision process with feedback of reward signals (Sutton et al., 1998) or richer signals like natural language (Cheng et al., 2024a; Chen et al., 2024). We consider an episodic setting. In each iteration, the agent interacts with the environment for multiple steps, receives feedback for each step (the feedback can be empty), and then updates its parameters at the end of the episode. To represent this structure as a computational graph, first we describe the interaction process of how observations and actions are generated. In Fig. 1, this is shown as a chain similar to a Markov decision process; notice there is an arrow going from action to the next observation via an operator denoted as $\Rightarrow$, which captures the causality. Then we apply the batchify operator $\oplus$ on the observations generated $\hat{o} = \oplus_{i=1}^T o_i$, where $T$ is the episode length, and similarly concatenates the feedback.

**Remark.** Using the right meta graph for a learning setup is important as it provides the learning context to the generative optimizer; otherwise objective misalignment can happen. For instance, if we desire a batch learning solution (i.e. a parameter that works well across a dataset of examples) but use the meta-graph for online learning in OPTO, we can get sub-par optimization results (for instance, an unstable parameter that is sensitive to the order in which individual examples are presented).

Similarly, if an agent's behavior has long-term consequences, then we should only change its behavior logic after an episode terminates. The separation between behavior logic and when/how to change them allows us to specify the right learning objective and allow the agent to learn the right behavior.

## 4 ATARI GAME PLAYING AGENTS THROUGH EPISODIC LEARNING

Game playing has been a central focus in reinforcement learning (Mnih et al., 2013; Silver et al., 2016; Brown & Sandholm, 2019). Recently, LLMs have demonstrated abilities to play long-horizon games such as Pokémon Blue (Karten et al., 2025; Anthropic, 2025). However, all of these successes utilize direct weight updates for the neural network, through training on collected in-game experience or massive pre-training on tutorials and forum posts. In this section, we want to demonstrate that, shockingly, with LLM as the optimizer and a correct learning template, we can learn a python program (not weights) that can play games that were typically mastered by neural networks.

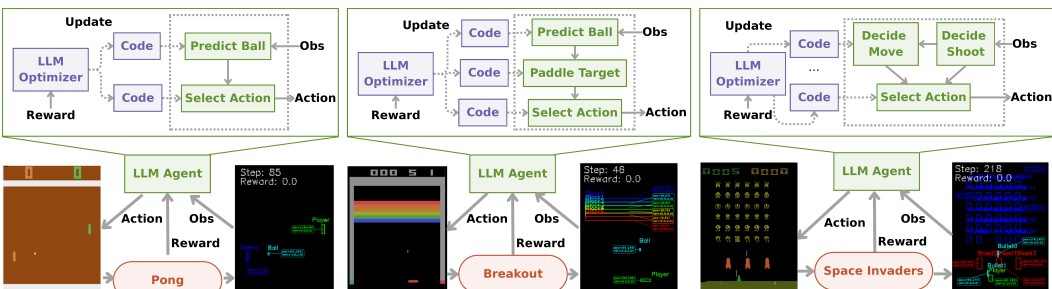

**Figure 2:** We show the workflow design of different decision-making program components for each Atari game agent. The LLM agent receives an object-centric dictionary of information of the game state and uses Python code to process and output an action.

The Arcade Learning Environment (ALE) of Atari games has remained an important benchmark for evaluating RL algorithms for training neural network-based policies (Mnih et al., 2013). ALE can be used to evaluate an RL algorithm in several ways: 1) The algorithm's learning efficiency both in terms of number of interactions with the environment and the overall wallclock time (Hessel et al., 2018); 2) The diverse set of environments allow the evaluation of generalization of learning (Lee et al., 2022).

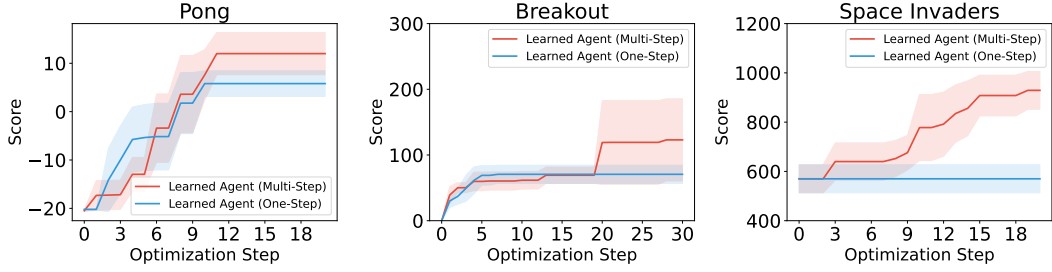

**Figure 3:** Performance of agent under different learning graphs (one-step vs multi-steps) across 5 trials. Episodic learning template concatenates workflow graph at each step to build an episodic learning graph that correctly specifies the temporal dependency in the agent's learning objective.

We use object-centric Atari Environments (OCAtari) (Delfosse et al., 2024) to parse the pixel-based observation from ALE to object-based representation. OCAtari provides the coordinates, size and velocity of the object on screen, game termination condition ("lives"), and current reward (see Figure A.9). We do not perform additional transformations to make the observation more readable.

**Workflow Design.** The agent is designed slightly differently for each game. The design decision is driven by a high-level modularization of the decision-making process. Both Pong and Breakout agents have `select_action` as the final component. They use `predict_ball_trajectory` as an intermediate step, where the prediction is provided to `select_action` to decide how the paddle can

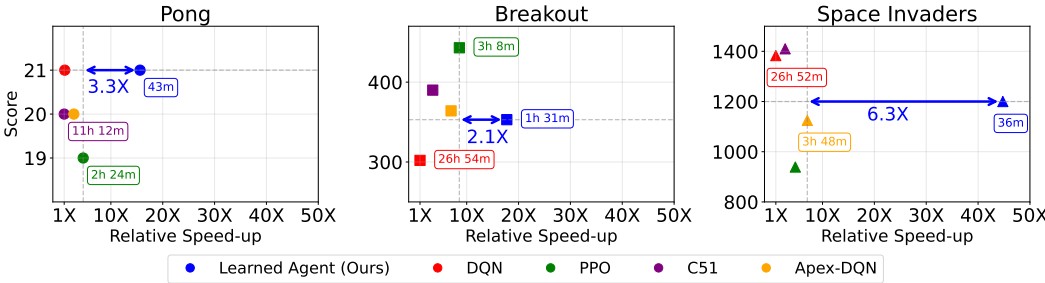

**Figure 4:** We show the relative speedup running the generative optimization process compared to traditional RL methods. Learned Agent result is the highest score it achieved in 5 trials. We report RL results from an open-source implementation of RL algorithms (Huang et al., 2022b) and publicly available experiment logs (Huang et al., 2022a). RL algorithms are trained with 8 parallel environment instances. Note that most recent Deep RL with 32 environment instances can earn score of 450 on Breakout in 33m, see Appendix F.5. The difference of scores (19-21) in Pong is not a meaningful difference and caused mostly by rounding.

be moved. For Breakout, we introduced a goal prediction component (`generate_paddle_target`) to strategically determine where the paddle should go to maximize the reward. For Space Invaders Agent, we simply have `decide_shoot` and `decide_movement` to decompose the decision space of choosing when to fire and when to move the game avatar.

**Learning Graph Design.** Due to the length of the context window for the LLM we use, we are only able to trace a fixed number of temporal steps. The number of steps is determined by the token spent representing the observation, the complexity of the agent design for the game, and the length of the solution. The training rollout is 300 steps for Breakout, 400 steps for Pong, and 10 steps for Space Invaders.

**Feedback Design.** We notice that only providing feedback based on the reward in the training rollout leads to performance plateaus, particularly in games where the game mechanism changes based on player progress. For example, in Breakout, the higher-value bricks in the upper rows deflect the ball at greater speeds, creating a distribution shift between the training context (primarily lower bricks) and the evaluation context (including higher bricks). This observation inspires two feedback design choices: 1) we provide staged feedback to instruct the model to pay attention to different game mechanisms or share high-level winning strategies; 2) we evaluate the performance of the agent with longer rollouts (up to 4000 steps) and use that reward as feedback to the generative optimizer.

**Results.** We find that even with sparse representation of game states and rewards in the form of trajectories, LLM optimizer (OptoPrime) demonstrate remarkable ability to infer game mechanics and environmental constraints from traced trajectories. While our approach provides docstrings that describe high-level game objectives and mechanics, we experiment with deliberately omitting specific implementation details like exact boundary coordinates or collision physics. Despite this, OptoPrime consistently infers these crucial details through analysis of the trajectory data. For example, in Breakout (see an example observation in Figure A.9), OptoPrime identifies the exact positions of the left wall ($x = 9$) and right wall ($x = 152$) by observing ball position and velocity changes across multiple steps. It correctly implements ball physics calculations including bounce mechanics without being explicitly told these details. This emergent understanding of game physics and boundaries demonstrates the LLM's ability to perform causal inference from sequential observations.

## 5 DATA SCIENCE AGENT WITH INTERACTIVE LEARNING GRAPH

The interactive learning setup can describe the learning objective for the majority of LLM agent benchmarks. The hallmark of these benchmarks is that even though an LLM agent needs to take multiple intermediate steps to complete a task, such step *does not* cause state transition in the environment that changes the reward the agent would receive. Even for benchmark that requires the agent to carry out multiple actions to successfully complete a task, the intermediate actions do not cause an internal, stateful change in the environment – the reward is often only associated with the final output of the agent (such as a customer response or an executable code).

MLAgentBench is a benchmark specifically designed to measure the effectiveness of machine learning agents in automating ML experimentation processes (Huang et al., 2023). There are different designs of the ML agents. The majority of agents created to solve this task have a primitive self-improvement loop, where the agent simply looks at its previous output and self-refine (Huang et al., 2023; Wang et al., 2024c; Chan et al., 2024). All tasks in MLAgentBench involves training a machine learning model, figuring out preprocessing data, feature selection, choosing hyperparameters of the model, and deciding training details. We chose two tabular tasks for the purpose of easy experimentation, as they consume the least amount of compute resources compared to large datasets.

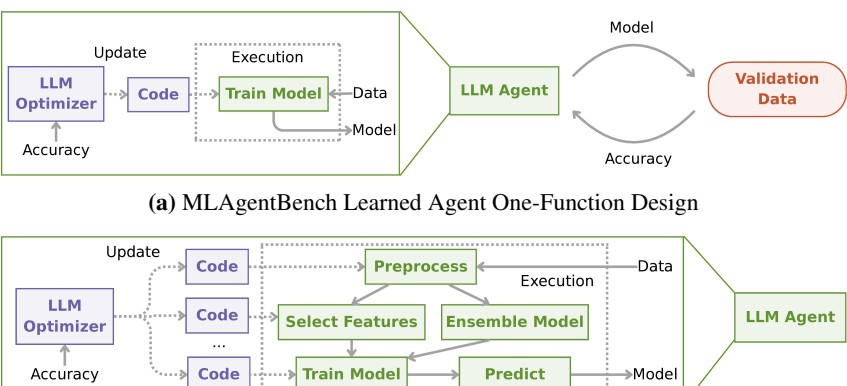

**(a)** MLAgentBench Learned Agent One-Function Design

**(b)** MLAgentBench Learned Agent Many-Function Design

Figure 5: **Agent Design for MLAgentBench**. We can design different agent workflow graphs for the learned agent to solve the task of training a machine learning model given a dataset.

**Workflow Design.** We design our agent to have specific components that are changable by the optimizer. That is, to set up a generative optimization process that is automatic, only with human engineering focused on the initial configurations. To design the agent's internal operation, we experiment with two different kinds of workflow design to highlight the influence of workflow design on the optimization outcome. One design asks the optimizer to program *one* code block that does everything, labeled as "One" in Table 1. The other design properly decomposes the model training tasks into five steps: `preprocess`, `select_features`, `create_ensemble_model`, `train_model`, and `predict`, labeled as "Many" in Table 1. We illustrate these two choices in Figure 5.

**Learning Design.** We use the OptoPrime as the generative optimizer (Cheng et al., 2024b). The entire graph is represented in the LLM context window and the LLM perform parameter update for all parameters all at once. We perform a train-validation split on the dataset to create a validation partition and use the task-specific metric on the validation dataset as the optimization objective (i.e., *maximize accuracy* or *minimize error*). We use the final learned agent's machine learning model to produce predictions on the hidden test set and submit to Kaggle website to compare against hidden ground truth. We *do not* use the Kaggle test score as the reward signal in the optimization loop.

**Feedback Design.** We apply fine-grained style feedback to the generative optimizer at different stages of validation accuracy (see Figure A.1). We additionally experimented with improvement style feedback where the model fails to train a machine learning model that has a higher validation accuracy than the previous step, we append an improvement suggestion to the feedback string.

**Results.** To make the comparison fair, we pre-downloaded the datasets for the Research Agent and made sure it could produce a machine learning model with valid test submission files for Kaggle (Huang et al., 2023). We track the average performance of the model produced by the both agents as well as the best result. After 20 optimization steps, we submit the model with the highest validation accuracy to the Kaggle competition to get the test score and leaderboard ranking. On both tasks, the gap between the Research Agent (Huang et al., 2023) and our learned agent is around 11.5%-22.4% on average, and the best machine learning model produced in the learned agent surpasses 86.6% of human submissions. Surprisingly, letting the optimizer continuously updating one function (block of code) is better for Housing Price, but not for Spaceship Titanic. This result highlights the importance of experimenting with different workflow designs.

|  |  | Housing Price RMSE (↓) | Spaceship Titanic Accuracy (↑) |
|---|---|---|---|
| ResearchAgent (Huang et al., 2023) | | | |
| Average | – | 0.149 | 78.17 |
| Best | – | 0.145 | 79.84 |
| Learned Agent (Ours) | | | |
| Average | One | **0.135** | 79.65 |
|  | Many | 0.147 | **79.69** |
| Best | One | **0.129** | 80.00 |
|  | Many | 0.141 | **80.43** |

**Table 1:** MLAgentBench Result. We run both agents 5 times and compute the average and best test score. Single and Many refer to a one-function vs many-functions workflow design for the agent (Figure 5). Both agents use the same underlying LLM (Claude Sonnet-3.5-v2).

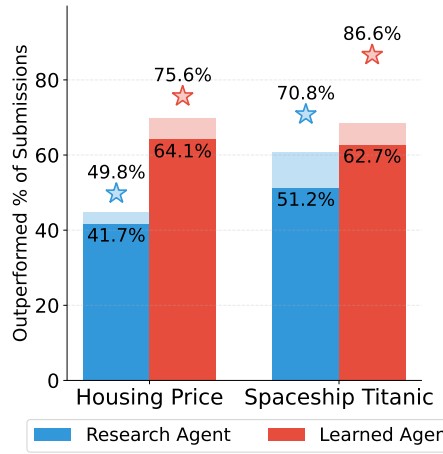

**Figure 6:** Leaderboard ranking for the best agent across design choices.

# 6 LANGUAGE UNDERSTANDING AGENT THROUGH BATCH LEARNING

From document processing to logical deductions, LLM agents are used for general language understanding tasks where the agent designer needs to write a pre/post-processing program as well as instructions/prompts to the LLM API call. The crucial learning context here is to allow the agent to write *one* prompt and a *fixed* program that generalizes to different kinds of questions and tasks. We explore the effect of setting the right learning context in BigBench Extra Hard (BBEH) (Kazemi et al., 2025).

**Agent Design.** We show the workflow design graph in Figure 7. This agent is made of two components, one is a *call llm* function that takes in the task query and a optimizable prompt. The other is an *answer extraction* function that parses the return from the LLM call.

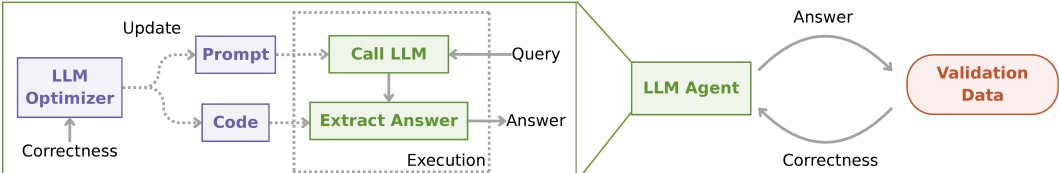

**Figure 7: Agent Design for BigBench Extra Hard (BBEH)**. Note that in here, we only show an interactive learning setup, where the agent graph only contains one query and receives one correctness feedback. In our experiment, we insert the execution graph into a template through batchify ⊕ operator to construct a batch learning graph over multiple queries, answers, and feedbacks.

**Learning Design** We apply the batch learning template to construct the learning graph for the agent. In order to apply this template, we sample a batch of inputs from the dataset $\mathcal{D}$. We roll out the workflow graph on each of the input, and by the end, we concatenate all the workflow graphs together to form the batch learning graph. We only use 20 example inputs for learning and the rest are holdout test set for evaluation. Batch learning graph helps agent understand how one shared program and prompt need to adapt to different kinds of inputs.

**Feedback Design** We provide feedback as a list of strings that contain whether the workflow's response for each input is correct or incorrect and revealing the solution to the optimizer when the answer is incorrect.

**Results.** We systematically change the batch learning graph's size (batch size) during the learning phase over 20 examples and measure how well the learned prompt and postprocessing code generalize to the unseen examples in the BBEH set. Surprisingly, even though generally presenting more than one example at a time (Batch Size > 1) gives better results (see Table 2), it seems that different batch sizes lead to different performances for different tasks. We also see different patterns of learning convergence on a 5-example validation set that we selected from the 20 examples (Figure 8). Larger

batch sizes allow the agent to learn faster but also plateau more quickly (Geometric Shapes). This highlights the benefit of constructing learning graphs – it exposes the hyperparameters of learning that agent designers must decide. We additionally report results on GSM-8K with the learned LangGraph agent in Section E and Table A.3.

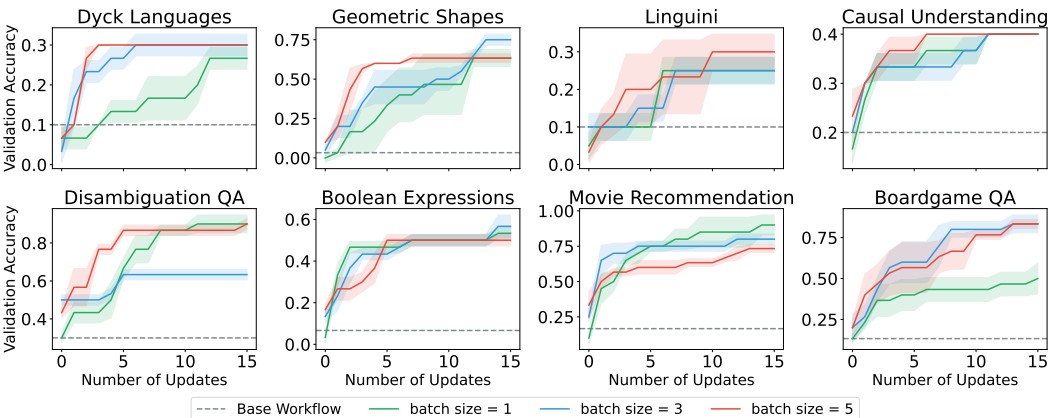

**Figure 8:** Performance of the learned workflow for each task across 3 trials. Each task starts with the same prompt and answer extraction code. Shaded area shows standard error. The training dataset size is fixed to 15 examples. Validation set has 10 examples.

| MiniBatch Size | Dyck Languages | Geometric Shapes | Linguini | Causal Understanding |
|---|---|---|---|---|
| Un-Optimized | $0.114 \pm 0.007$ | $0.074 \pm 0.003$ | $0.183 \pm 0.010$ | $0.114 \pm 0.005$ |
| Batch Size=1 | $0.183 \pm 0.049$ | $0.343 \pm 0.039$ | $0.149 \pm 0.024$ | $0.375 \pm 0.146$ |
| Batch Size=3 | $0.063 \pm 0.010$ | $\mathbf{0.389} \pm 0.040$ | $\mathbf{0.234} \pm 0.012$ | $0.408 \pm 0.097$ |
| Batch Size=5 | $\mathbf{0.190} \pm 0.031$ | $0.200 \pm 0.099$ | $0.170 \pm 0.030$ | $\mathbf{0.531} \pm 0.018$ |

**Table 2:** Holdout Test Set Performance for BBEH tasks. Bold indicates best accuracy per column; standard error is shown in smaller gray text. The test dataset excludes examples used for train and val, and usually includes 175 examples. The full table for 8 tasks are in Appendix A.2.

## 7  CONCLUSION AND LIMITATION

We demonstrate generative optimization on computational graph is a powerful new paradigm for agent learning. We identify common misalignment issues in practice and provide constructive guidelines to address them. With these insights, we demonstrate successful agent learning results on a wide range of problems (GSM8K, BBEH, MLAgentBench, and Atari games) across interactive, batch, and reinforcement learning scenarios. These experimental results push the boundary of problems where generative optimization has been applied in the literature and provide strong evidence that generative optimization can be the key to the next breakthrough of agent learning and an effective method to leverage inference time compute to find optimal solution automatically.

However, we should also highlight that our current results have limitations. Although we solved the objective misalignment with a principled reductionist approach, our current recommendations on agent and feedback design are still heuristic-driven. We also notice that the optimization process can be unstable. The guideline can largely mitigate the issue, but efforts are still required to configure the initial condition and optimization procedure correctly. All of these warrant future research to explore paths to create a fully automated, goal-driven, *generalist* agent.

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

## A  Large Language Model Access Card

The experiments were conducted during the period of February 2025 to April 2025. The model used was Anthropic's Claude Sonnet 3.5v2, and the exact model name was "anthropic.claude-3-5-sonnet-20241022-v2:0". All of the agents, including the baseline agents from other papers (ResearchAgent (Huang et al., 2023), PAL agent (Gao et al., 2023a), Self-Refine agent (Madaan et al., 2023)) were all rerun using the same model endpoint as our learned agent during the same period of time.

## B  Discussion on Other Important Factors for Agent Learning

### B.1  Specifying Agent Behavior Through Workflows

The same agent can be described by different workflow graphs, but these graphs may have different optimization properties. We discuss factors that affect optimization difficulty.

**Modularization.** Breaking down the reasoning process from a monolithic block into multiple smaller blocks has proven useful for complex reasoning tasks (Zhou et al., 2023). It remains an empirical question to what extent one should modularize, but the guiding principle here is to decompose a monolithic process into a computational graph with smaller operators.

**Parametrization.** There are some parts of the workflow that do not need much design exploration. For example, loading in a text file given a file name would always result in similar code (even with error handling). Designing an optimizable workflow requires engineers to think carefully and parametrize the part of their workflow where the optimal solution is not known a priori to them, where exploring new parameters is valuable.

**Initialization.** Similar to the research on neural network initialization (e.g. (Glorot & Bengio, 2010)), the optimizable functions contain initial code and docstrings designated by engineers. We found that a workflow that involves operators with ambiguous docstrings is difficult to optimize via generative optimization. We advise using initialization that conveys a clear (desired) behaviors of operators used in the graph.

### B.2  Guiding Agent Learning Process Through Effective Feedback

Prior works have studied the importance of providing feedback to the optimization process (Chen et al., 2023; Nie et al., 2023; Wei et al., 2024). We highlight and summarize a few useful types of implementations below.

**Fine-grained Feedback.** The simplest form of feedback is to include the correct/incorrect information or the numerical reward in text. However, other feedback designs have also proven to be useful. In Section 4, we show that we can have *staged* feedback, where a different feedback is used when the agent reaches different reward regions (Table A.6). This allows flexibility on how to guide the generative optimizer to search the solution space. Another way is to identify trigger keywords from environment information (such as a system profiler) and retrieve corresponding pre-written feedback (Wei et al., 2024).

**Suggestive Feedback.** The best type of feedback tells the generative optimizer exactly how to change the parameter or output – it should be actionable. If it is not possible to know this information, a suggestion should still be made with proper degree of suggestions in the phrasing. Earlier this was referred to as "directional" feedback (Nie et al., 2023) and later Wei et al. (2024) showed suggestive feedback allowed the optimizer to find better solutions than explanation-based feedback.

## C  MLAGENTBENCH DETAILS

### C.1  AGENT DESIGN DETAILS

The ML agent shares a similar design for both tasks with modular components for different steps of the machine learning pipeline.

### C.2  FEEDBACK DESIGN DETAILS

We provided task-specific feedback instructions when the agent reaches different performance level. We show the feedback template in Figure A.1. The {SUGGESTION} block if filled by the suggestive feedback in Table A.1.

```
1 Epoch {epoch}/20:
2
3 Accuracy: {val_accuracy:.4f}
4 F1: {val_f1:.4f}
5 Precision: {val_precision:.4f}
6 Recall: {val_recall:.4f}.
7
8 {SUGGESTION}
```

**Figure A.1:** The feedback template used for the ML Agent for Spaceship-Titanic.

| Accuracy | Suggestive Feedback |
| --- | --- |
| Val F1 $< 0.5$ | "Model performance is poor. Try better feature engineering and preprocessing." |
| $0.5 \leq$ Val F1 $< 0.7$ | "Model is showing promise but needs improvement. Consider class balancing techniques." |
| $0.7 \leq$ Val F1 $< 0.8$ | "Model is performing well. Fine-tune hyperparameters for further improvements." |
| Val F1 $\geq 0.8$ | "Excellent performance! Focus on preventing overfitting." |

**Table A.1:** Staged suggestive feedback for the ML agent at different accuracy levels for the Spaceship-Titanic task.

### C.3  LLM AGENT LEARNING RESULTS

We perform a training and validation split outside of the agent and only pass the training set as input to the agent. This is due to the fact that generative optimization requires an optimization signal. Kaggle does not permit more than 5 submissions on the test set per day, therefore, we do not use the test set as our optimization signal. We randomly split the training data, providing 80% to the agent –

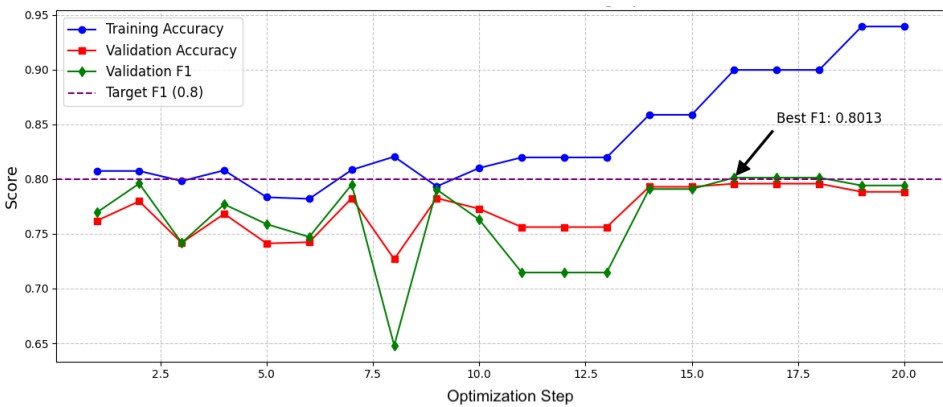

**Figure A.2:** We show the training/validation accuracy and F1 score from machine learning model outputted by the ML Agent after each optimization step. Note that the machine learning model outputted could be trained internally for hundreds of epochs. The x-axis describes the number of optimization steps in the generative optimizer to update the parameter of the ML Agent.

the agent is allowed to further split that data into train and validation. We use the 20% as our test set to evaluate the agent's machine learning model's performance. We show the learning progress of one trial run in Figure A.2. The x-axis of this figure shows the optimization steps. Although on a cursory glance, this graph seems to be depicting typical model overfitting behavior as training accuracy goes up and validation accuracy goes down as optimization continues, it is however not the case. At each optimization step, the agent is producing a fully trained machine learning model using the training dataset, with however many numbers of training iterations it chooses. This figure shows the phenomenon of meta-overfitting, where the generative optimizer updates the agent to choose hyperparameters, training procedures of the model that overfits the training set, even though the feedback reward comes purely from the validation performance.

## C.4 EFFECT OF WORKFLOW DESIGN

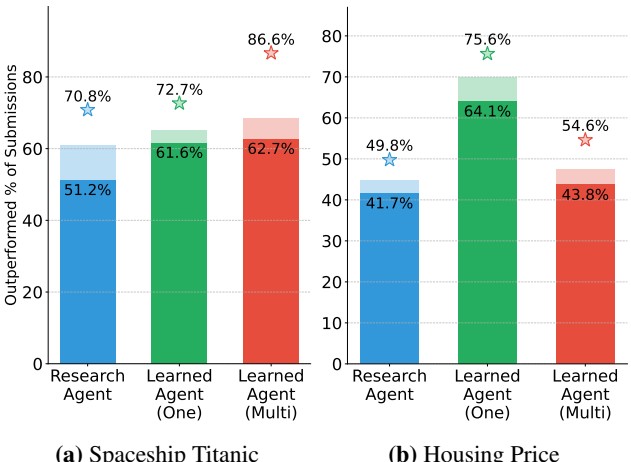

(a) Spaceship Titanic      (b) Housing Price

**Figure A.3:** Leaderboard performance metrics for learned model submissions.

## C.5 EXAMPLES OF THE LEARNED ML AGENT

We provide initial code (with docstrings) for each game and the final learned code. For Spaceship-Titanic Agent, initial code is in Figure A.10, A.11, A.12, A.13, A.14. Some of the functions are not heavily updated, but we showcase the learned final functions in Figure. Essentially, the generative

optimizer chose to tune the numbers (hyperparameters) of machine learning models and ensemble method.

# D    BATCH LEARNING AGENT DETAILS

| MiniBatch | Dyck Languages | Geometric Shapes | Linguini | Causal Understanding |
|---|---|---|---|---|
| Un-Optimized | $0.114 \pm 0.007$ | $0.074 \pm 0.003$ | $0.183 \pm 0.010$ | $0.114 \pm 0.005$ |
| Batch=1 | $0.183 \pm 0.049$ | $0.343 \pm 0.039$ | $0.149 \pm 0.024$ | $0.375 \pm 0.146$ |
| Batch=3 | $0.063 \pm 0.010$ | $\mathbf{0.389} \pm 0.040$ | $\mathbf{0.234} \pm 0.012$ | $0.408 \pm 0.097$ |
| Batch=5 | $\mathbf{0.190} \pm 0.031$ | $0.200 \pm 0.099$ | $0.170 \pm 0.030$ | $\mathbf{0.531} \pm 0.018$ |

| MiniBatch | Disambiguation QA | Boolean Expressions | Movie Recommendation | Boardgame QA |
|---|---|---|---|---|
| Un-Optimized | $0.358 \pm 0.013$ | $0.076 \pm 0.009$ | $0.238 \pm 0.007$ | $\mathbf{0.371} \pm 0.003$ |
| Batch=1 | $\mathbf{0.537} \pm 0.036$ | $0.177 \pm 0.005$ | $\mathbf{0.889} \pm 0.038$ | $0.341 \pm 0.032$ |
| Batch=3 | $0.295 \pm 0.091$ | $\mathbf{0.238} \pm 0.006$ | $0.683 \pm 0.119$ | $0.278 \pm 0.009$ |
| Batch=5 | $0.526 \pm 0.035$ | $0.154 \pm 0.034$ | $0.810 \pm 0.016$ | $0.276 \pm 0.007$ |

**Table A.2:** Performance across tasks and batching strategies. Best test accuracy per task is bolded; standard error is shown in smaller gray text. For Boardgame QA, we observe meta-overfitting: the learned workflow had strong validation scores but failed to generalize to test examples. Base model is Claude Sonnet-3.5-v2.

# E    BATCH LEARNING LANGGRAPH AGENT DETAILS

The emergence of LLM multi-agent frameworks (Wu et al., 2024; LangChain, 2024) allow static programs to have dynamic behaviors enabled by LLMs. Here we apply generative optimization to such frameworks to enable an agentic workflow to improve itself.

**Agent Design.**    We used LangGraph (LangChain, 2024) to implement two popular LLM agent designs. The first one is program-aided language model (PAL) (Gao et al., 2023a). This agent design consists of two components: first, it tries to produce a Python program conditioned on a prompt and the input. Then it executes the program to get the final answer. We learn both of these components. The second agent is a self-refine agent (Madaan et al., 2023), where the agent would use a function to solve the problem, verify its solution, and if the solution is wrong, it will try to refine the solution until it passes the verification step.

**Learning Design** We use LangGraph to build the workflow but use OptoPrime (Cheng et al., 2024b) as the optimizer to learn all the mentioned modules. At each iteration, execution traces from a minibatch of examples are captured, evaluated in terms of feedback, and aggregated. Feedback is concatenated into an aggregated feedback, which is then processed by the optimizer. Before implementing batch learning, optimizing by example would overfit, leading to over-specialized improvements that failed to generalize.

**Feedback Design** We provide feedback as string with templates for both correct and incorrect responses, revealing the solution to the optimizer when the answer is incorrect.

**Results.** Empirical results in Table A.3 confirm the efficacy of the generative optimization framework. We evaluate on GSM8K (Kazemi et al., 2025) and BBEH (Kazemi et al., 2025). For GSM8K, we use the same train/validation/test split as used in DSPy (Khattab et al., 2024). For BBEH, we chose two tasks as representative examples to verify our pipeline. The baselines are both PAL and self-refine agent implemented with good initial starter code and working prompts. However, the learned agent (both code and prompts are learned) performs much better on GSM8K, increasing the performance from 78.9% to 93.4%. For BBEH, the initial agent was not able to output answers with a valid format without few-shot examples. In this zero-shot setup, the optimizer is able to find good prompts and valid code to ensure the produced answer is correct.

| | GSM8K | BBEH Causal Understanding | BoardgameQA |
|---|---|---|---|
| PAL Agent (Gao et al., 2023b) | 78.9 | 5.0 | 5.0 |
| Learned PAL Agent (Ours) | **93.4** | **42.5** | **33.0** |
| Self-Refine Agent (Madaan et al., 2023) | 78.2 | 0.0 | 0.0 |
| Learned Self-Refine Agent (Ours) | **86.8** | **44.0** | **32.5** |

**Table A.3:** Comparison of baseline LLM agent design and their optimized design on GSM8K and BBEH.

### E.1 AGENT DESIGN DETAILS

The PAL (Program-Aided Language Model) agent (Gao et al., 2023a) is designed to have two functions: parse_problem and execute_code. parse_problem makes a call to the LLM with the following prompt "Read the problem and output a Python expression to compute the answer and store it into 'result' variable. Problem: {}". The execute_code uses Python's "exec" function to execute the program written by parse_problem. Both functions are updated by the generative optimizer.

The self-refine agent (Madaan et al., 2023) is designed to have three functions: solve_problem, verify_solution, and refine_solution. All three are LLM calling functions with prompt strings defined within the function.

- solve_problem: "Solve the following problem step by step and give the final answer: {question}. Solution:"

- verify_solution: "You are a math expert. Verify the solution below for correctness. Problem: {question}. Solution and Answer: {solution}. Is the answer correct? If not, explain the error."

- refine_solution: "The previous answer was found to be incorrect. {verification_feedback}. Please solve the problem again correctly: {question}. Correct Solution:"

Each function takes the LangGraph's state dictionary as input and was added as nodes to LangGraph's StateGraph for execution. The self-refinement loop is controlled by LangGraph's conditional routing strategy.

### E.2 FEEDBACK DESIGN DETAILS

The feedback given to the optimizer when the answer is correct is "ANSWER IS CORRECT/SUCCESS" and when the answer is wrong, we reveal the reference solution since there is a training phase for the agent: "WRONG ANSWER / FAILED - your answer: {answer} vs. good answer: {solution}".

## F ATARI GAME DETAILS

### F.1 GAME SETUP

The training configuration is reported in Table A.4 and the environment setup is reported in Table A.5. The Atari Gym offers many wrappers to help with learning. Atari environments by default uses frameskip (repeat actions) to reduce the horizon length and use sticky action probability to randomly repeat the previous action with given probability. Both were designed to enable better training for the deep neural network. In our experiment, we found that not using sticky action results in better optimization of the model.

We generate data on-the-fly for Atari games using object-centric Atari Environments (OCAtari) (Delfosse et al., 2024), a wrapper for the Gymnasium API (Towers et al., 2024) that provides object-centric representation of the game screen at each

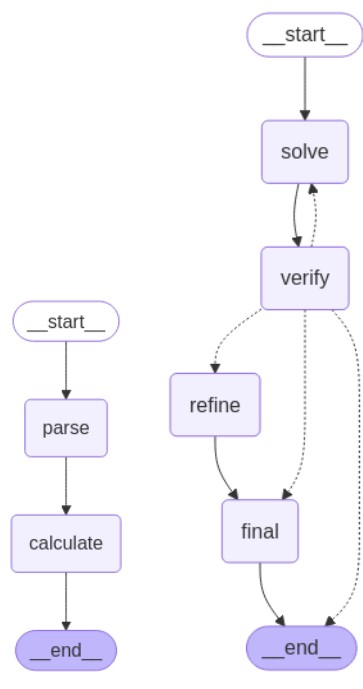

**(a)** PAL     **(b)** Self-refine agent

**Figure A.4:** The LangGraph visualization of workflow.

| Parameter | Value |
|---|---|
| Environment name | {env}-NoFrameskip-v4 |
| Action repeat (frameskip) | 4 |
| Sticky action probability | 0.0 |
| Optimization iterations | 30 |
| Rollout length | 15/300/400 steps |
| Memory size (optimizer context) | 5 |
| Evaluation episode length | ~4000 steps |
| LLM optimizer | OptoPrime |
| LLM Backend | Claude-3.5 Sonnet-20241022-v2:0 |
| Access Date | 3/20/2025 |

**Table A.4:** Atari Gym environment and training configurations

| Parameter | Breakout | Pong | Space Invaders |
|---|---|---|---|
| Rollout horizon | 300 steps | 400 steps | 15 steps |
| Action space | LEFT/RIGHT/ NOOP | UP/DOWN/ NOOP | LEFT/RIGHT FIRE/NOOP |
| Env special mechanics | Auto-fire on life loss | None | Fire cooldown |

**Table A.5:** Atari game-specific experiment configurations

timestep. For instance, for the game Pong, OCAtari returns the position $(x, y)$, size (width, height), and velocity $(dx, dy)$ of the player paddle, ball, and enemy paddle. This representation abstracts away from raw pixel inputs, providing the LLM optimizer and our agent with structured state information that facilitates targeted improvements to the agent's prediction and action selection. The actual input observation to the agent is shown in Figure A.9, and an annotated screen through OCAtari can be seen in Figure A.5, A.6, and A.7.

**Pong**    In Pong, the player controls a paddle on the right side of the screen to deflect the ball into the enemy's goal. The player scores a point if the enemy misses the ball. The game ends when one side scores 21 points.

**Breakout**    In Breakout, the player moves a bottom paddle horizontally to deflect a ball that scores against brick walls upon contact. The brick wall consists of six rows of different colored bricks, with higher bricks worth more points. Hitting higher bricks would deflect the ball faster, increasing the difficulty in catching the ball. The player wins after scoring 864 points. The player loses one life when failing to catch the ball and the ball moves out of range. The player has five lives in total.

**Space Invaders**    In Breakout, the player moves a bottom paddle horizontally to deflect a ball that scores against brick walls upon contact. The brick wall consists of six rows of different colored bricks, with higher bricks worth more points. Hitting higher bricks would deflect the ball faster, increasing the difficulty in catching the ball. The player wins after scoring 864 points. The player loses one life when failing to catch the ball and the ball moves out of range. The player has five lives in total.

## F.2   FEEDBACK DESIGN DETAILS

We provided game-specific feedback instructions when the agent reaches different reward regions.

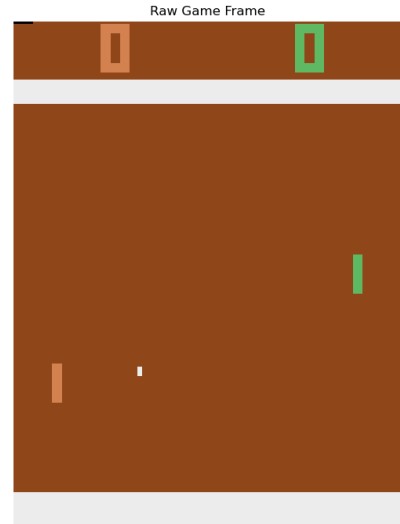
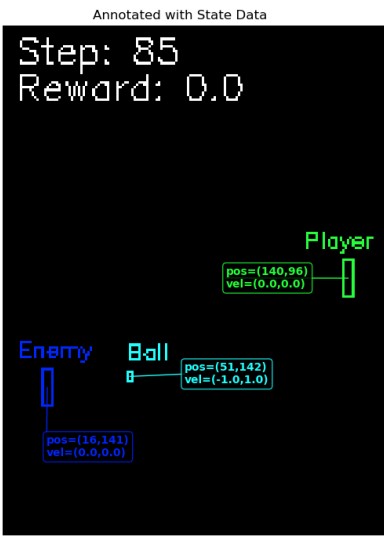

**Figure A.5: Pong**: An annotated screenshot to show how OCAtari (Delfosse et al., 2024) translates objects from pixels to obejcts with annotations.

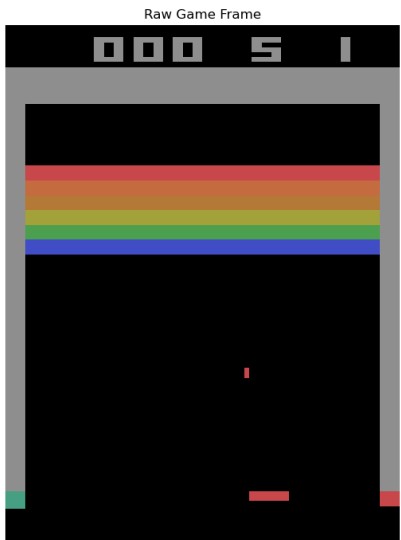
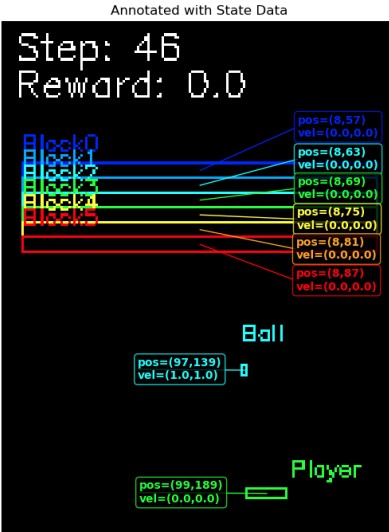

**Figure A.6: Breakout**: An annotated screenshot to show how OCAtari (Delfosse et al., 2024) translates objects from pixels to obejcts with annotations.

### F.3 AGENT DESIGN DETAILS

**Pong**   In order to succeed at Pong, the agent should accurately predict where the ball will intersect with the player's paddle plane, accounting for bounces off the top and bottom walls. Thus, we adapt our base agent architecture to focus on ball trajectory prediction and paddle positioning (predict_ball_trajectory() and select_action()). We initialize predict_ball_trajectory() to return the current $y$ coordinate of the ball and select_action() to return a random action of UP or DOWN. In the docstring, we provide detailed description of the game screen, including screen dimensions and paddle positions. We show the initialized agent in Figure A.16 and the optimized agent in Figure A.17.

**Breakout**   Breakout has a similar emphasis of considering wall boucing but with a focus on brick targeting. Like Pong, we adapt our base agent architecture to focus on predicting the trajectory of the ball (predict_ball_trajectory()), but also prioritizing hitting bricks with higher scores (generate_paddle_target())

Raw Game Frame

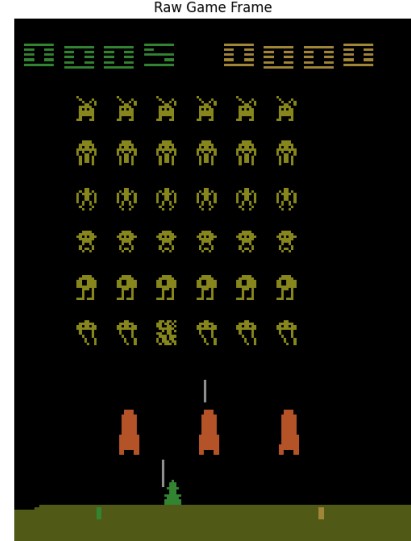

Annotated with State Data

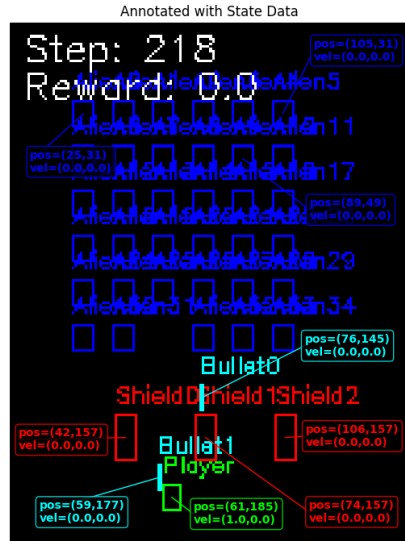

**Figure A.7: Space Invaders**: An annotated screenshot to show how OCAtari (Delfosse et al., 2024) translates objects from pixels to obejcts with annotations.

| Performance Level | Feedback |
|---|---|
| High (Reward ≥ 19) | "Good job! You're close to winning the game! You're scoring 20 points against the opponent, only 1 points short of winning." |
| Medium (0 < Reward < 19) | "Keep it up! You're scoring 12 points against the opponent but you are still 9 points from winning the game. Try improving paddle positioning to prevent opponent scoring." |
| Low (Reward ≤ 0) | "Your score is −5 points. Try to improve paddle positioning to prevent opponent scoring." |

**Table A.6:** Staged feedback for the Pong agent at different performance levels

and selecting paddle action based on the analysis (`select_paddle_action()`). We initialize both `predict_ball_trajectory()` and `generate_paddle_target()` to return None, and `select_paddle_action()` to move the paddle `LEFT` or `RIGHT` by comparing the paddle location to the target position generated by `generate_paddle_target()`, which is None upon initialization. In the docstring, we describe the game screen, such as locations of left and right wall, but we leave the exact location out for the LLM to infer based on the traced trajectory. We also describe the point system of the brick wall and some generic strategic considerations (without telling the agent how to implement these strategies). We show the initialized agent in Figure A.18, A.19 and the optimized agent in Figure A.20, A.21.

**Space Invaders** For Space Invaders, we adapt our base agent architecture to into two tasks of deciding whether to shoot (`decide_shoot()`) and deciding where to move (`decide_move()`), and finally combining the two decisions in (`combine_actions()`). We initialize `decide_shoot()` and `decide_movement()` to return random actions, and `combine_actions()` map the outputs of the previous two functions to the correct action. In the docstring, we describe the game setup and the presence of shield objects. We show the initialized agent in Figure A.22, A.23 and the learned agent in Figure A.24.

F.4 LLM AGENT LEARNING RESULT

| Performance Level | Example Feedback |
|---|---|
| High (Reward $\geq$ 300) | "Good job! You're close to winning the game! You're scoring 320 points against the opponent, try ensuring you return the ball, only 30 points short of winning." |
| Medium (0 < Reward < 300) | "Keep it up! You're scoring 50 points against the opponent but you are still 300 points from winning the game. Try improving paddle positioning to return the ball and avoid losing lives." |
| Low (Reward $\leq$ 0) | "Your score is -5 points. Try to improve paddle positioning to return the ball and avoid losing lives." |

**Table A.7:** Staged feedback for the Breakout agent at different performance levels

| Performance Level | Feedback |
|---|---|
| High (Reward $\geq$ 1000) | "Great job! You're performing well with an average score of 1005. Try to score more even more points" |
| Medium (500 < Reward < 1000) | "Good progress! Your average score is 570. Focus on better timing for shooting and avoiding enemy projectiles." |
| Low (Reward $\leq$ 500) | "Your average score is 270. Try to improve your strategy for shooting aliens and dodging projectiles." |

**Table A.8:** Staged feedback for the Space Invaders agent at different performance levels

### F.5 DEEP RL RESULT

Due to a large variation in how people report Atari game results and the fact that many state-of-the-art deep RL models are not released as open-source, the numbers we reported in Table A.9 are from CleanRL report (Huang et al., 2022b), the published ICLR blog post (Huang et al., 2022a) and the experiment log[1]. In terms of runtime, we directly compute the time from the Weights & Biases log. For Breakout and Space Invaders, the agent performances were continuously improving, so we reported the duration of the full experiment run. For Pong, the RL policy plateaued before the experiment finished, so we found the time step where the policy achieved the highest performance reliably and computed training time starting from the launch of the experiment to that time step.

It is worth noting that we reporeted the Deep RL results with 8 parallel environment instances in Table A.9. However, there are faster implementations of Deep RL training on Atari games. For example, Apex-DQN (Horgan et al., 2018) would train the actor and critic model separately in a truly asynchronous fashion, resulting in massive reduction of training time. EnvPool is a C++-based batched environment pool that enabled fast sampling and interaction with the game environment. All of these changes enabled faster learning. For example, on Breakout, with 32 to 64 parallel environments, Advantage Actor-Critic (A2C) can learn a high performing policy in 33m 19s[2]. However, Trace only uses 1 environment instance and has not gone through any special speed-related algorithm/hardware optimization.

### F.6 EXAMPLES OF THE LEARNED ATARI AGENT

We provide initial code (with docstrings) for each game and the final learned code. For Pong Agent, initial code is in Figure A.16, and final agent in Figure A.17. For Breakout agent, initial code is in

---

[1] https://wandb.ai/cleanrl/cleanrl.benchmark/reports/Atari--VmlldzoxMTExNTI

[2] https://wandb.ai/costa-huang/cleanRL/reports/Breakout-v5--VmlldzoxNDI1MTIx

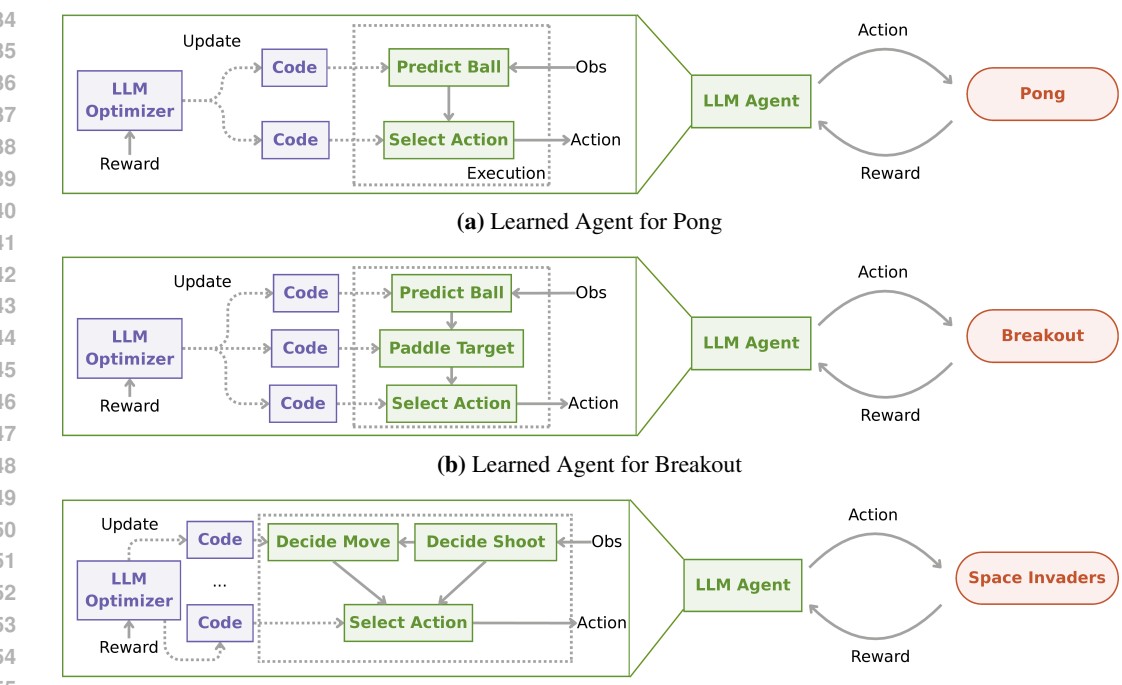

**(a)** Learned Agent for Pong

**(b)** Learned Agent for Breakout

**(c)** Learned Agent for Space Invaders

**Figure A.8: Agent Design for Atari Games**. We can design different agent workflow graphs for the learned agent to achieve high scores in three Atari games.

```
TracedEnv.step.step16 = {
    'Player': {'x': 99, 'y': 189, 'w': 16, 'h': 4, 'dx': 0, 'dy': 0},
    'Ball': {'x': 7, 'y': 193, 'w': 2, 'h': 4, 'dx': -4, 'dy': 4},
    'RB': [{'x': 8, 'y': 57, 'w': 144, 'h': 6}],
    'OB': [{'x': 8, 'y': 63, 'w': 144, 'h': 6}],
    'YB': [{'x': 8, 'y': 69, 'w': 144, 'h': 6}],
    'GB': [{'x': 8, 'y': 75, 'w': 144, 'h': 6}],
    'AB': [{'x': 8, 'y': 81, 'w': 144, 'h': 6}],
    'BB': [{'x': 8, 'y': 87, 'w': 144, 'h': 6}],
    'lives': 5,
    'reward': 0.0
}
```

**Figure A.9:** Example of a single traced step from Breakout

Figure A.18, A.19 and final agent in Figure A.20, A.21. For Space Invaders agent, initial code is in Figure A.22, A.23 and the final agent in Figure A.24.

| Game | Learned Agent | DQN (Time) | PPO (Time) | Human |
|---|---|---|---|---|
| Pong | **21** (43m) | 20 (10h 6m) | 19 (2h 24m) | 14.59 |
| Breakout | **353** (1h 31m) | 302 (26h 54m) | 443 (3h 8m) | 30.47 |
| Space Invaders | **1200** (36m) | 1383 (26h 52m) | 939 (5h 39m) | 1668.67 |

**Table A.9:** Comparison of Algorithm Performance on Atari Games with Time. Due to high variations of numbers reported by different papers, we report results from an open-source implementation of RL algorithms (Huang et al., 2022b) and publicly available experiment logs (Huang et al., 2022a). RL algorithms are trained with 8 parallel environment instances. Our agent is trained on 1 environment instance. Note that highly optimized Deep RL with 32 environment instances can reach ~450 on Breakout in 33m, see Appendix F.5.

# G   LARGE LANGUAGE MODEL USE FOR WRITING

A small amount of paragraphs have been polished by GPT-5. The process is – the author wrote the sentence or paragraph first and then send into the LLM with the prompt of "Polish the following writing and correct the grammar mistakes." LLM was never used to directly produce a paragraph without an original human-written input. The LLM assistance was only used to enhance the calrity and readability of the paragraph only.

# H   EXAMPLES OF LEARNED AGENT

## H.1   ML AGENT

## H.2   ATARI GAME AGENTS

```python
import trace

@trace.model
class SpaceshipTitanicPipeline(Module):

    def __call__(self, x, y=None, test_data=None):
        processed_data = self.preprocess(x)
        selected_features = self.select_features(processed_data)

        if y is not None:
            ensemble = self.ensemble_model(selected_features, processed_data)
            model = self.train_model(ensemble, selected_features, processed_data, y)
            if test_data is not None:
                processed_test_data = self.preprocess(test_data)
                filtered_test_data = self.filter_features(selected_features,
        processed_test_data)
                return self.predict(model, filtered_test_data)
            filtered_data = self.filter_features(selected_features, processed_data)
            return self.predict(model, filtered_data)
        else:
            ensemble = self.ensemble_model(selected_features, processed_data)
            processed_test_data = self.preprocess(x)
            filtered_test_data = self.filter_features(selected_features, processed_test_data)
            model = self.train_model(ensemble, selected_features, processed_data,
        pd.Series([False] * len(processed_data)))
            return self.predict(model, filtered_test_data)

    def filter_features(self, selected_features, data):
        return data[selected_features]

    @trace.bundle(trainable=True)
    def preprocess(self, data):
        """
        Preprocessing Steps (some examples on how you could do this, however you can use
        your own method if it works better):
        1. Missing Value Handling:
            - Numerical features: Intelligent imputation (median, mean, or 0)
            - Categorical features: Mode filling or meaningful defaults
            - Outlier detection and treatment

        2. Feature Engineering:
            - Passenger ID parsing:
                * Extract group and individual identifiers
                * Create group-related features
            - Cabin information extraction:
                * Deck identification
                * Cabin number parsing
                * Side (port/starboard) classification
            - Name feature parsing:
                * Title extraction
                * Potential family relationship inference

        3. Advanced Feature Creation:
            - Family size computation
            - Total and relative spending calculations
            - Amenity usage patterns
            - Spatial features (cabin location metrics)

        4. Categorical Variable Handling:
            - One-hot encoding
            - Label encoding
            - Embedding techniques for high-cardinality features

        5. Numerical Feature Transformation:
            - Scaling (StandardScaler, MinMaxScaler)
            - Skewness correction (log, square root, Box-Cox)
            - Normalization techniques

        Args:
            data (pd.DataFrame): Raw input dataset

        Returns:
            pd.DataFrame: Preprocessed dataset with engineered features
        """
        return data
```

**Figure A.10:** Initial code for Spaceship-Titanic ML Agent (Part 1). Docstrings are generated by ChatGPT and then edited by humans.

```python
@trace.model
class SpaceshipTitanicPipeline(Module):
    # (continued from above)
    @trace.bundle(trainable=True)
    def select_features(self, processed_data):
        """
        Select the most relevant features for predicting whether passengers were transported.

        Selection Methodology (some examples on how you could do this, however you can use
        your own method if it works better):
        1. Statistical Feature Importance:
            - Correlation analysis
            - Mutual information
            - Chi-squared tests
            - Model-based feature importance

        2. Feature Weighting Criteria:
            - Predictive power for transportation status
            - Domain-specific relevance
            - Minimal multicollinearity
            - Computational efficiency

        3. Key Feature Categories:
            - Demographic Signals
            - Travel Characteristics
            - Economic Indicators

        4. Selection Mechanism:
            - Probabilistic feature selection
            - Dynamic weight adjustment
            - Prevent overfitting through selective inclusion

        Args:
            processed_data (pd.DataFrame): Preprocessed dataset

        Returns:
            list: Optimally selected feature names with associated weights
        """
        all_features_with_weights = {col: 0.5 for col in processed_data.columns}

        available_features = {k: v for k, v in all_features_with_weights.items() if k in
    processed_data.columns}

        feature_names = list(available_features.keys())
        feature_weights = list(available_features.values())

        num_features = min(len(feature_names), int(len(feature_names) * 0.8))

        selected_features = np.random.choice(
            feature_names,
            size=num_features,
            replace=False,
            p=[w/sum(feature_weights) for w in feature_weights]
        ).tolist()

        selected_features = [f for f in selected_features if f in processed_data.columns]

        return selected_features
```

**Figure A.11:** Initial code for Spaceship-Titanic ML Agent (Part 2). Docstrings are generated by ChatGPT and then edited by humans.

```python
@trace.model
class SpaceshipTitanicPipeline(Module):
    # (continued from above)
    @trace.bundle(trainable=True)
    def ensemble_model(self, features, data):
        """
        Create an ensemble model for predicting passenger transport status.

        Ensemble Strategy (some examples on how you could do this, however you can use your
        own method if it works better):
        1. Model Diversity:
            - Tree-based models (Random Forest, Gradient Boosting)
            - Linear models (Logistic Regression variants)
            - Support Vector Machines
            - Probabilistic classifiers

        2. Ensemble Techniques:
            - Voting, boosting, bagging, stacking
            - Stacking with meta-learners
            - Weighted model combination
            - Regularization-aware model selection

        3. Hyperparameter Optimization:
            - Cross-validated parameter tuning
            - Regularization strength balancing
            - Learning rate and depth control
            - Subsample and feature sampling strategies

        4. Computational Considerations:
            - Computational complexity management
            - Memory-efficient model design
            - Scalable ensemble construction

        Args:
            features (list): Selected feature names
            data (pd.DataFrame): Processed dataset

        Returns:
            sklearn Classifier: Configured ensemble model ready for training
        """
        models = [
            ('rf', RandomForestClassifier(n_estimators=150, max_depth=10,
        min_samples_split=5,  min_samples_leaf=4, max_features='sqrt',random_state=42)),
            ('gbr', GradientBoostingClassifier(n_estimators=200, learning_rate=0.03,
        max_depth=3, subsample=0.8, min_samples_split=5, random_state=42)),
            ('xgb', XGBClassifier(n_estimators=200, learning_rate=0.03, max_depth=3,
        colsample_bytree=0.6, subsample=0.8, reg_alpha=0.1, reg_lambda=1.0, gamma=1,
        random_state=42)),
            ('lasso', LogisticRegression(penalty='l1', C=0.1, random_state=42)),
            ('ridge', LogisticRegression(penalty='l2', C=20.0, random_state=42)),
            ('elastic', LogisticRegression(penalty='elasticnet', C=0.1, l1_ratio=0.8,
        random_state=42))
        ]

        ensemble = VotingClassifier(
            estimators=models,
            voting='soft',
            weights=[2, 3, 3, 2, 1, 1]
        )

        return
```

**Figure A.12:** Initial code for Spaceship-Titanic ML Agent (Part 3). Docstrings are generated by ChatGPT and then edited by humans.

```
1  @trace.model
2  class SpaceshipTitanicPipeline(Module):
3      # (continued from above)
4      @trace.bundle(trainable=True)
5      def train_model(self, ensemble_model, features, data, results):
6          """
7          Train machine learning models to predict whether passengers were transported.
8
9          Training Methodology (some examples on how you could do this, however you can use
           your own method if it works better):
10             1. Data Preparation:
11                 - Feature subset preparation
12                 - Cross-validation splitting
13                 - Stratified sampling
14
15             2. Class Imbalance Handling:
16                 - Weighted loss functions
17                 - SMOTE oversampling
18                 - Synthetic data generation
19                 - Class-aware regularization
20
21             3. Regularization Techniques:
22                 - L1/L2 penalty integration
23                 - Dropout-like regularization
24                 - Early stopping mechanisms
25                 - Gradient clipping
26
27             4. Training Optimization:
28                 - Adaptive learning rates
29                 - Ensemble member performance tracking
30                 - Dynamic weight adjustment
31                 - Prediction confidence calibration
32
33             Args:
34                 ensemble_model: Configured ensemble model
35                 features (list): Selected feature names
36                 data (pd.DataFrame): Processed training dataset
37                 results (pd.Series): Training labels
38
39             Returns:
40                 Trained ensemble model optimized for passenger transportation prediction
41             """
42             return
```

**Figure A.13:** Initial code for Spaceship-Titanic ML Agent (Part 4). Docstrings are generated by ChatGPT and then edited by humans.

```python
@trace.model
class SpaceshipTitanicPipeline(Module):
    # (continued from above)
    @trace.bundle(trainable=True)
    def predict(self, model, data):
        """
        Make predictions on whether passengers were transported.

        Prediction Workflow (some examples on how you could do this, however you can use
        your own method if it works better):
        1. Probabilistic Prediction:
            - Soft classification probabilities
            - Confidence-based thresholding
            - Ensemble prediction aggregation

        2. Post-processing Techniques:
            - Calibration curves
            - Probability scaling
            - Ensemble diversity preservation

        3. Output Formatting:
            - Binary classification output
            - Kaggle submission compatibility
            - Interpretable prediction format

        4. Prediction Quality Assessment:
            - Uncertainty quantification
            - Prediction reliability scoring
            - Anomaly detection

        Args:
            model (VotingClassifier): Trained ensemble model
            data (pd.DataFrame): Processed test dataset

        Returns:
            np.ndarray: Binary predictions for passenger transportation status
        """
        predictions = model.predict(data)
        predictions = np.array(predictions, dtype=bool)
        return predictions
```

**Figure A.14:** Initial code for Spaceship-Titanic ML Agent (Part 5). Docstrings are generated by ChatGPT and then edited by humans.

```python
import trace

@trace.model
class SpaceshipTitanicPipeline(Module):

    @trace.bundle(trainable=True)
    def preprocess(self, data):
        """(same as before, skipped to save space)"""
        # Create a copy to avoid modifying original data
        # Handle missing values in numeric columns
        numeric_columns = ["Age", "RoomService", "FoodCourt", "ShoppingMall",
            "Spa", "VRDeck"]
        for col in numeric_columns:
            processed_data[col] = processed_data[col].fillna(processed_data[col].median())

        # Handle boolean/categorical columns
        processed_data["VIP"] = processed_data["VIP"].fillna(False)
        processed_data["CryoSleep"] = processed_data["CryoSleep"].fillna(False)

        # Convert HomePlanet to numeric using label encoding
        if "HomePlanet" in processed_data.columns:
            processed_data["HomePlanet"] = processed_data["HomePlanet"].fillna("Unknown")
            planet_map = {"Earth": 0, "Europa": 1, "Mars": 2, "Unknown": 3}
            processed_data["HomePlanet"] = processed_data["HomePlanet"].map(planet_map)

        # (skipped some code)

        # Age-related features
        processed_data["Age"] = processed_data["Age"].fillna(processed_data["Age"].median())
        processed_data["AgeGroup"] = pd.qcut(
            processed_data["Age"], q=6, labels=[0, 1, 2, 3, 4, 5]
        ).astype(int)

        # Interaction features
        processed_data["CryoSleepVIP"] = processed_data["CryoSleep"].astype(int) *
        processed_data["VIP"].astype(int)
        processed_data["SpendingPerAge"] = processed_data["TotalSpending"] /
        processed_data["Age"].clip(lower=1)
        processed_data["HasSpent"] = (processed_data["TotalSpending"] > 0).astype(int)
        processed_data["SpendingVariety"] = (processed_data[spending_columns] >
        0).sum(axis=1)

        # ... standard scaling, dropping columns, etc.

        # Final check for NaN values
        processed_data = processed_data.fillna(0)
        return processed_data
```

**Figure A.15:** Final learned code for Spaceship-Titanic ML Agent (Part 1). Docstrings are generated by ChatGPT and then edited by humans.

```
1 import trace
2
3 @trace.model
4 class Policy(Module):
5     def __call__(self, obs):
6         predicted_ball_y = self.predict_ball_trajectory(obs)
7         action = self.select_action(predicted_ball_y, obs)
8         return action
9
10    @trace.bundle(trainable=True)
11    def predict_ball_trajectory(self, obs):
12        """
13        Predict the y-coordinate where the ball will intersect with the player's paddle by
        calculating its trajectory,
14        using ball's (x, y) and (dx, dy) and accounting for bounces off the top and bottom
        walls.
15
16        Game Setup:
17        - Screen dimensions: The game screen has boundaries where the ball bounces
18          - Top boundary: approximately y=30
19          - Bottom boundary: approximately y=190
20        - Paddle positions:
21          - Player paddle: right side of screen (x = 140)
22          - Enemy paddle: left side of screen (x = 16)
23
24        Args:
25            obs (dict): Dictionary containing object states for "Player", "Ball", and
        "Enemy".
26                    Each object has position (x,y), size (w,h), and velocity (dx,dy).
27
28        Returns:
29            float: Predicted y-coordinate where the ball will intersect the player's paddle
        plane.
30                Returns None if ball position cannot be determined.
31
32        """
33        if 'Ball' in obs:
34            return obs['Ball'].get("y", None)
35        return None
36
37    @trace.bundle(trainable=True)
38    def select_action(self, predicted_ball_y, obs):
39        '''
40        Select the optimal action to move player paddle by comparing current player position
        and predicted_ball_y.
41
42        IMPORTANT! Movement Logic:
43        - If the player paddle's y position is GREATER than predicted_ball_y: Move DOWN
        (action 2)
44            (because the paddle needs to move downward to meet the ball)
45        - If the player paddle's y position is LESS than predicted_ball_y: Move UP (action 3)
46            (because the paddle needs to move upward to meet the ball)
47        - If the player paddle is already aligned with predicted_ball_y: NOOP (action 0)
48            (to stabilize the paddle when it's in position)
49        Ensure stable movement to avoid missing the ball when close by.
50
51        Args:
52            predicted_ball_y (float): predicted y coordinate of the ball or None
53            obs(dict): Dictionary of current game state, mapping keys ("Player", "Ball",
        "Enemy") to values (dictionary of keys ('x', 'y', 'w', 'h', 'dx', 'dy') to integer
        values)
54        Returns:
55            int: 0 for NOOP, 2 for DOWN, 3 for UP
56        '''
57
58        if predicted_ball_y is not None and 'Player' in obs:
59            return random.choice([2, 3])
60        return 0
```

**Figure A.16:** Initial code for Pong Agent.

```python
import trace

@trace.model
class Policy(Module):
    def __call__(self, obs):
        predicted_ball_y = self.predict_ball_trajectory(obs)
        action = self.select_action(predicted_ball_y, obs)
        return action

    @trace.bundle(trainable=True)
    def predict_ball_trajectory(self, obs):
        """(same as before, skipped to save space)"""
        if 'Ball' in obs:
            ball = obs['Ball']
            # If ball moving away from player, return None
            if ball.get('dx', 0) < 0:
                return None

            # Calculate time to reach paddle
            paddle_x = 140
            ball_x = ball.get('x', 0)
            ball_dx = ball.get('dx', 0)
            if ball_dx == 0:
                return ball.get('y', None)

            time_to_paddle = (paddle_x - ball_x) / ball_dx

            # Calculate predicted y position with improved accuracy
            ball_y = ball.get('y', 0)
            ball_dy = ball.get('dy', 0)
            predicted_y = ball_y + ball_dy * time_to_paddle

            # Account for bounces with improved accuracy
            num_bounces = 0
            while predicted_y < 30 or predicted_y > 190:
                if predicted_y < 30:
                    predicted_y = 30 + (30 - predicted_y)
                if predicted_y > 190:
                    predicted_y = 190 - (predicted_y - 190)
                num_bounces += 1
                if num_bounces > 4:  # Limit bounce calculations
                    break

            return predicted_y
        return None

    @trace.bundle(trainable=True)
    def select_action(self, predicted_ball_y, obs):
        '''(same as before, skipped to save space)'''
        if predicted_ball_y is not None and 'Player' in obs:
            # Calculate center of paddle
            paddle_center = obs['Player']['y'] + obs['Player']['h']/2

            # Increase margin and add dynamic adjustment based on ball distance
            base_margin = 4
            if 'Ball' in obs:
                ball_x = obs['Ball'].get('x', 0)
                dist_factor = (140 - ball_x) / 140  # Normalized distance factor
                margin = base_margin * (1 + dist_factor)  # Larger margin when ball is far

                # Add momentum-based adjustment
                if obs['Ball'].get('dx', 0) > 0:
                    ball_dy = obs['Ball'].get('dy', 0)
                    # Scale adjustment based on distance
                    predicted_ball_y += ball_dy * dist_factor
            else:
                margin = base_margin

            # More aggressive movement thresholds
            if paddle_center > predicted_ball_y + margin:
                return 2  # Move down
            elif paddle_center < predicted_ball_y - margin:
                return 3  # Move up
            return 0  # Stay in position
        return 0
```

**Figure A.17:** Final learned code for Pong Agent.

```python
@trace.model
class Policy(Module):
    def __call__(self, obs):
        pre_ball_x = self.predict_ball_trajectory(obs)
        target_paddle_pos = self.generate_paddle_target(pre_ball_x, obs)
        action = self.select_paddle_action(target_paddle_pos, obs)
        return action

    @trace.bundle(trainable=True)
    def predict_ball_trajectory(self, obs):
        """
        Predict the x-coordinate where the ball will intersect with the player's paddle by
        calculating its trajectory,
        using ball's (x, y) and (dx, dy) and accounting for bounces off the right and left
        walls.

        Game setup:
        - Screen dimensions: The game screen has left and right walls and brick wall where
        the ball bounces
            - Left wall: x=9
            - Right wall: x=152
        - Paddle positions:
            - Player paddle: bottom of screen (y=189)
        - Ball speed:
            - Ball deflects from higher-scoring bricks would have a higher speed and is harder
        to catch.
            - The paddle would deflect the ball at different angles depending on where the ball
        lands on the paddle

        Args:
            obs (dict): Dictionary containing object states for "Player", "Ball", and blocks
        "{color}B" (color in [R/O/Y/G/A/B]).
                        Each object has position (x,y), size (w,h), and velocity (dx,dy).
        Returns:
            float: Predicted x-coordinate where the ball will intersect the player's paddle
        plane.
                    Returns None if ball position cannot be determined.
        """
        if 'Ball' not in obs:
            return None

    @trace.bundle(trainable=True)
    def generate_paddle_target(self, pre_ball_x, obs):
        """
        Calculate the optimal x coordinate to move the paddle to catch the ball (at
        predicted_ball_x)
        and deflect the ball to hit bricks with higher scores in the brick wall.

        Logic:
        - Prioritize returning the ball when the ball is coming down (positive dy)
        - The brick wall consists of 6 vertically stacked rows from top to bottom:
            - Row 1 (top): Red bricks (7 pts)
            - Row 2: Orange (7 pts)
            - Row 3: Yellow (4 pts)
            - Row 4: Green (4 pts)
            - Row 5: Aqua (1 pt)
            - Row 6 (bottom): Blue (1 pt)
          - Strategic considerations:
            - Breaking lower bricks can create paths to reach higher-value bricks above
            - Creating vertical tunnels through the brick wall is valuable as it allows
              the ball to reach and bounce between high-scoring bricks at the top
            - Balance between safely returning the ball and creating/utilizing tunnels
              to access high-value bricks
        - Ball speed increases when hitting higher bricks, making it harder to catch

        Args:
            pre_ball_x (float): predicted x coordinate of the ball intersecting with the
        paddle or None
            obs (dict): Dictionary containing object states for "Player", "Ball", and blocks
        "{color}B" (color in [R/O/Y/G/A/B]).
                        Each object has position (x,y), size (w,h), and velocity (dx,dy).
        Returns:
            float: Predicted x-coordinate to move the paddle to.
                Returns None if ball position cannot be determined.
        """
        if pre_ball_x is None or 'Ball' not in obs:
            return None
        return None
```

**Figure A.18:** Initial code for Breakout Agent (Part 1).

```
1  import trace
2
3  @trace.model
4  class Policy(Module):
5
6      # (continued from above)
7
8      @trace.bundle(trainable=True)
9      def select_paddle_action(self, target_paddle_pos, obs):
10         """
11         Select the optimal action to move player paddle by comparing current player position
           and target_paddle_pos.
12
13         Movement Logic:
14         - If the player paddle's center position is GREATER than target_paddle_pos: Move
           LEFT (action 3)
15         - If the player paddle's center position is LESS than target_paddle_pos: Move RIGHT
           (action 2)
16         - If the player paddle is already aligned with target_paddle_pos: NOOP (action 0)
17           (to stabilize the paddle when it's in position)
18         Ensure stable movement to avoid missing the ball when close by.
19
20         Args:
21             target_paddle_pos (float): predicted x coordinate of the position to best
           position the paddle to catch the ball,
22             and hit the ball to break brick wall.
23             obs (dict): Dictionary containing object states for "Player", "Ball", and blocks
           "{color}B" (color in [R/O/Y/G/A/B]).
24             Each object has position (x,y), size (w,h), and velocity (dx,dy).
25         Returns:
26             int: 0 for NOOP, 2 for RIGHT, 3 for LEFT
27         """
28         if target_paddle_pos is None or 'Player' not in obs:
29             return 0
30
31         paddle = obs['Player']
32         paddle_x = paddle['x']
33         paddle_w = paddle['w']
34         paddle_center = paddle_x + (paddle_w / 2)
35
36         # Add deadzone to avoid oscillation
37         deadzone = 2
38         if abs(paddle_center - target_paddle_pos) < deadzone:
39             return 0  # NOOP if close enough
40         elif paddle_center > target_paddle_pos:
41             return 3  # LEFT
42         else:
43             return 2  # RIGHT
```

**Figure A.19:** Initial code for Breakout Agent (Part 2).

```python
@trace.model
class Policy(Module):

    def __call__(self, obs):
        pre_ball_x = self.predict_ball_trajectory(obs)
        target_paddle_pos = self.generate_paddle_target(pre_ball_x, obs)
        action = self.select_paddle_action(target_paddle_pos, obs)
        return action

    @trace.bundle(trainable=True)
    def predict_ball_trajectory(self, obs):
        """(same as before, skipped to save space)"""
        if pre_ball_x is None or 'Ball' not in obs or 'Player' not in obs:
            return None

        ball = obs['Ball']
        paddle = obs['Player']

        # Default to centering paddle on predicted ball position
        target_x = pre_ball_x

        # Adjust paddle position based on current ball direction and brick locations
        if ball['dy'] > 0:  # Ball moving down
            if ball['y'] < 120:  # Ball in upper half - aim for tunnels to high bricks
                # Look for gaps in brick rows to target
                high_brick_x = None
                for color in ['RB', 'OB']:  # Check red and orange rows
                    if color in obs:
                        bricks = obs[color]
                        if len(bricks) > 0:
                            brick = bricks[0]
                            high_brick_x = brick['x'] + (brick['w'] / 2)
                            break

                if high_brick_x is not None:
                    # Adjust paddle to deflect ball toward high-value bricks
                    if ball['x'] < high_brick_x:
                        target_x = pre_ball_x - 4  # Hit ball on right side
                    else:
                        target_x = pre_ball_x + 4  # Hit ball on left side

        # Ensure target is within screen bounds
        paddle_half_width = paddle['w'] / 2
        target_x = max(9 + paddle_half_width, min(152 - paddle_half_width, target_x))

        return target_x
```

**Figure A.20:** Learned code for Breakout Agent (Part 1).

```python
@trace.model
class Policy(Module):

    # (continued from above)

    @trace.bundle(trainable=True)
    def generate_paddle_target(self, pre_ball_x, obs):
        """(same as before, skipped to save space)"""
        if 'Ball' not in obs:
            return None

        ball = obs['Ball']
        ball_x = ball['x']
        ball_y = ball['y']
        ball_dx = ball['dx']
        ball_dy = ball['dy']

        # If ball is not moving, can't predict
        if ball_dx == 0 and ball_dy == 0:
            return None

        # Calculate time to reach paddle
        paddle_y = 189  # Player paddle y-position
        if ball_dy != 0:  # Avoid division by zero
            time_to_paddle = (paddle_y - ball_y) / ball_dy
            if time_to_paddle > 0:  # Only predict if ball is moving toward paddle
                # Account for wall bounces
                x = ball_x + (ball_dx * time_to_paddle)
                num_bounces = 0
                while x < 9 or x > 152:
                    if x < 9:
                        x = 9 + (9 - x)  # Reflect off left wall
                    if x > 152:
                        x = 152 - (x - 152)  # Reflect off right wall
                    num_bounces += 1
                    if num_bounces > 10:  # Prevent infinite bounces
                        break
                return x
        return None

    @trace.bundle(trainable=True)
    def select_paddle_action(self, target_paddle_pos, obs):
        """(same as before, skipped to save space)"""
        if target_paddle_pos is None or 'Player' not in obs:
            return 0

        paddle = obs['Player']
        paddle_x = paddle['x']
        paddle_w = paddle['w']
        paddle_center = paddle_x + (paddle_w / 2)

        # Add deadzone to avoid oscillation
        deadzone = 2
        if abs(paddle_center - target_paddle_pos) < deadzone:
            return 0  # NOOP if close enough
        elif paddle_center > target_paddle_pos:
            return 3  # LEFT
        else:
            return 2  # RIGHT
```

**Figure A.21:** Learned code for Breakout Agent (Part 2).

```
1  @trace.model
2  class Policy(Module):
3
4      def __call__(self, obs):
5          shoot_decision = self.decide_shoot(obs)
6          move_decision = self.decide_movement(obs)
7          return self.combine_actions(shoot_decision, move_decision)
8
9      @trace.bundle(trainable=True)
10     def decide_shoot(self, obs):
11         '''
12         Decide whether to shoot based on enemy positions and existing projectiles.
13
14         Args:
15             obs (dict): Game state observation containing object states for "Player",
         "Shield0", "Shield1", "Alien0", "Alien1", etc.
16             Each object has position (x,y), size (w,h), and velocity (dx,dy).
17             Player bullets have negative dy velocity and alien bullets have positive dy
         velocity
18
19         Strategy tips:
20         - You can only have one missile at a time
21         - Try to shoot when aliens are aligned with your ship
22         - Prioritize shooting at lower aliens as they're closer to you
23         - Consider the movement of aliens when deciding to shoot
24
25         Returns:
26             bool: True if should shoot, False otherwise
27         '''
28
29         # There can only be one player bullet on the field at a time
30         # Check for player bullets (which have negative dy velocity)
31         for key, obj in obs.items():
32             if key.startswith('Bullet') and obj.get('dy', 0) < 0:
33                 return False
34
35         return random.choice([True, False])
36
37     @trace.bundle(trainable=True)
38     def decide_movement(self, obs):
39         '''
40         Decide movement direction based on enemy positions and projectiles.
41
42         Args:
43             obs (dict): Game state observation containing object states for "Player",
         "Shield0", "Shield1", "Alien0", "Alien1", etc.
44             Each object has position (x,y), size (w,h), and velocity (dx,dy).
45             Player bullets have negative dy velocity and alien bullets have positive dy
         velocity
46
47         Strategy tips:
48         - Move to dodge enemy projectiles
49         - Position yourself under aliens to shoot them
50         - Stay away from the edges of the screen
51         - Consider moving toward areas with more aliens to increase score
52
53         Returns:
54             int: -1 for left, 1 for right, 0 for no movement
55         '''
56
57         player = obs['Player']
58
59         return random.choice([-1,0,1])
```

**Figure A.22:** Initial code for Space Invaders Agent (Part 1).

```
1  @trace.model
2  class Policy(Module):
3
4      # (continued from above)
5
6      @trace.bundle(trainable=True)
7      def combine_actions(self, shoot, movement):
8          '''
9          Combine shooting and movement decisions into final action.
10
11         Args:
12             shoot (bool): Whether to shoot
13             movement (int): Movement direction
14
15         Action mapping:
16         - 0: NOOP (no operation)
17         - 1: FIRE (shoot without moving)
18         - 2: RIGHT (move right without shooting)
19         - 3: LEFT (move left without shooting)
20         - 4: RIGHT+FIRE (move right while shooting)
21         - 5: LEFT+FIRE (move left while shooting)
22
23         Returns:
24             int: Final action (0: NOOP, 1: FIRE, 2: RIGHT, 3: LEFT, 4: RIGHT+FIRE, 5:
         LEFT+FIRE)
25         '''
26
27         if shoot and movement > 0:
28             return 4  # RIGHT+FIRE
29         elif shoot and movement < 0:
30             return 5  # LEFT+FIRE
31         elif shoot:
32             return 1  # FIRE
33         elif movement > 0:
34             return 2  # RIGHT
35         elif movement < 0:
36             return 3  # LEFT
37         return 0  # NOOP
```

**Figure A.23:** Initial code for Space Invaders Agent (Part 2).

```python
@trace.model
class Policy(Module):

    def __call__(self, obs):
        shoot_decision = self.decide_shoot(obs)
        move_decision = self.decide_movement(obs)
        return self.combine_actions(shoot_decision, move_decision)

    @trace.bundle(trainable=True)
    def decide_shoot(self, obs):
        """(same as before , skipped to save space)"""

        # There can only be one player bullet on the field at a time
        # Check for player bullets (which have negative dy velocity)
        for key, obj in obs.items():
            if key.startswith('Bullet') and obj.get('dy', 0) < 0:
                return False

        player = obs['Player']
        for key, obj in obs.items():
            if key.startswith('Alien'):
                # Check if alien is aligned with player (within 5 pixels)
                if abs(obj['x'] - player['x']) < 5:
                    # Prioritize lower aliens (higher y value)
                    if obj['y'] > 60:  # Adjust this threshold as needed
                        return True
        return False

    @trace.bundle(trainable=True)
    def decide_movement(self, obs):
        """(same as before , skipped to save space)"""
        player = obs['Player']
        move = 0
        threat_left = 0
        threat_right = 0
        aliens_left = 0
        aliens_right = 0

        for key, obj in obs.items():
            if key.startswith('Alien'):
                if obj['x'] < player['x']:
                    aliens_left += 1
                else:
                    aliens_right += 1
            elif key.startswith('Bullet') and obj['dy'] > 0:  # Enemy bullet
                if obj['x'] < player['x']:
                    threat_left += 1
                else:
                    threat_right += 1

        # Move away from threats
        if threat_left > threat_right:
            move = 1
        elif threat_right > threat_left:
            move = -1
        # If no immediate threat, move towards more aliens
        elif aliens_left > aliens_right:
            move = -1
        elif aliens_right > aliens_left:
            move = 1

        return move

    @trace.bundle(trainable=True)
    def combine_actions(self, shoot, movement):
        """(same as before , skipped to save space)"""
        if shoot and movement > 0:
            return 4  # RIGHT+FIRE
        elif shoot and movement < 0:
            return 5  # LEFT+FIRE
        elif shoot:
            return 1  # FIRE
        elif movement > 0:
            return 2  # RIGHT
        elif movement < 0:
            return 3  # LEFT
        return 0  # NOOP
```

**Figure A.24:** Learned code for Space Invaders Agent.

