# OpenReview forum: "Building Learning Context For Autonomous Agents Through Generative Optimization"
_ICLR.cc/2026/Conference — ICLR 2026 Conference Withdrawn Submission_

### Official Review · Reviewer_isEt · 2025-10-23

**Soundness:** 2
**Presentation:** 3
**Contribution:** 1
**Rating:** 2
**Confidence:** 3

**Summary:**

The paper is focusing on the application of Generative Optimization to address the under-specification problem in autonomous agent learning. It proposes a fix by adding template-based "meta-graphs" that guide the LLM optimizer.

**Strengths:**

Main strength is in the introduction of a learning context meta-graph that guides an LLM, acting as an optimizer, to iteratively refine an agent's executable code (workflow) using rich, non-differentiable feedback.

**Weaknesses:**

The primary goal is to build an optimized agent computational graph from repeatable workflow graphs (basically several standard agent execution steps: generate, rerank, summarize), which is identical to reconstructing/reorganizing the agent context on the given task. There is actually no significant or superior difference compared to other agent learning frameworks [1, 2, 3, 4]. This work could be more insightful by comparing with other agent learning frameworks.

Batch learning or Episodic learning template, in essence, are two different ways to construct the problem feedback based on the generated workflow graph, one is parallel, the other is sequential. The paper provides these two different ways to construct the meta-graph, but in the experiments, but not provide any further investigation/discussion of the reasoning for designing these two types of graphs, nor they perform ablations on the performance of agent learning using two different graphs on the same problem.

[1] Ma, X., Lin, C., Zhang, Y., Tresp, V., & Ma, Y. (2025). Agentic Neural Networks: Self-Evolving Multi-Agent Systems via Textual Backpropagation. arXiv preprint arXiv:2506.09046.

[2] Wei, A., Nie, A., Teixeira, T. S., Yadav, R., Lee, W., Wang, K., & Aiken, A. (2024). Improving Parallel Program Performance with LLM Optimizers via Agent-System Interfaces. arXiv preprint arXiv:2410.15625.

[3] Yuan, S., Chen, Z., Xi, Z., Ye, J., Du, Z., & Chen, J. (2025). Agent-R: Training Language Model Agents to Reflect via Iterative Self-Training. arXiv preprint arXiv:2501.11425.

[4] Yang, Y., Kang, S., Lee, J., Lee, D., Yun, S. Y., & Lee, K. (2025). Automated Skill Discovery for Language Agents through Exploration and Iterative Feedback. arXiv preprint arXiv:2506.04287.


Minor:

In abstract, “We investigate three types of software engineering problems …", shouldn’t it be four types?

Diagram needs to be replotted; they are currently not readable, see Fig. 1, Fig. 2, Figure A.4, etc.

**Questions:**

As the paper suggested, this proposed framework can generalize to different domains. While in this work, the paper is focusing mainly on engineering problems. Can you expand the application of this framework to some open-ended tasks, such solving research problems?

The experiments conducted in this work are generally have a clear/explicit task objective. How can a generative computational graph be effectively structured and trained when the task objective is unknown or highly unstructured ?

---

### Official Review · Reviewer_8Nik · 2025-10-29

**Soundness:** 2
**Presentation:** 2
**Contribution:** 2
**Rating:** 4
**Confidence:** 3

**Summary:**

The paper argues that using an LLM as an optimizer over an LLM-based agent’s workflow parameters often fails because the learning context is under-specified. Therefore, it proposes specifying a meta-graph, which is constructed by inserting a workflow graph into one of three learning templates (interactive, batch, episodic) via operators so that the optimizer’s objective is aligned with what the agent designer actually wants. Comprehensive empirical validation across software engineering, reasoning, and control tasks, highlighting substantial performance and efficiency gains over deep RL baselines.

**Strengths:**

- The paper well addresses the under-specification problem in existing LLM-based generative optimization frameworks.
- The work is implemented and evaluated the framework across three distinct learning regimes, including interactive (MLAgentBench), batch (GSM8K, BBEH), and episodic (Atari).
- The paper is well-organized and pedagogically clear.

**Weaknesses:**

- There is no ablation that directly compares performance across identical workflows and feedback signals with and without the meta-graph structure.
- Missing baselines which formalize LLM-as-optimizer paradigms [1]. They share the same optimization philosophy but differ in graph specification or update strategy, excluding them makes it difficult to attribute improvements uniquely to context specification.
- Although the paper claims training-time savings, it does not specify how “training time” is measured. Moreover, the paper acknowledges that optimization is sometimes unstable, but does not quantify variance across runs.
- Results rely on carefully crafted “staged” or “suggestive” feedback, yet the sensitivity of performance to phrasing or feedback noise is untested.

---

[1] Zhou, W., Ou, Y., Ding, S., et al. (2024). Symbolic learning enables self-evolving agents. arXiv preprint arXiv:2406.18532.

**Questions:**

- Did the authors conduct any ablation experiments where the workflow and feedback oracle were held constant and only the meta-graph template was varied?
- How sensitive is performance to the phrasing or structure of feedback? Could the authors quantify robustness to prompt perturbations or randomized feedback?
- The work mentions instability and potential “meta-overfitting” but provide only qualitative discussion. What mechanisms or criteria do you use to detect divergence or overfitting during generative optimization?

---

### Official Review · Reviewer_CKF7 · 2025-10-31

**Soundness:** 3
**Presentation:** 2
**Contribution:** 2
**Rating:** 4
**Confidence:** 2

**Summary:**

This paper frames agent learning as a generative optimization problem by representing the agent’s behavior and its update process as computational graphs (meta-graphs). It proposes explicit learning contexts (interactive, batch, episodic) to facilitate self-improvement of LLM-based agents across domains.

**Strengths:**

* Provides a unified and general framework for agent learning that works even without differentiability.
* Demonstrates versatility across tasks (Atari, MLAgentBench, QA) with notable performance gains.

**Weaknesses:**

* The method is overly conceptual and abstract, relying heavily on heuristics that vary across application domains (e.g., feedback design).
* The framework has low reproducibility, as entire process of optimization heavily depends on LLMs.
* Lack of comprison to baselines. There are several relevant baselines that automatively designing agent framwork like ADAS (Automated Design of Agentic Systems). It would be nice if authors clarify distinction of the proposed framework compared to such previous approach.
* Writing is quite hard to follow. Especially in the method section, it would improve readability if the authors included more concrete examples and clearer references to illustrative figures when explaining the proposed framework.

**Questions:**

* How much cost does the generative optimization take? It would be better if you can discuss more on optimization steps and costs depending on each task setup.

---

### Official Review · Reviewer_R7SN · 2025-11-01

**Soundness:** 2
**Presentation:** 1
**Contribution:** 2
**Rating:** 2
**Confidence:** 4

**Summary:**

This paper proposes an improved method for LLM-based agent optimization by constructing a meta-graph via templates. This introduces the appropriate learning context to the LLM optimizer, addressing the under-specification of agent learning and enabling more robust behavior aligned with the designer’s objectives.

**Strengths:**

- The results on Atari seem promising.
- Experiments span three distinct domains (data science, computer security, and game playing)

**Weaknesses:**

- The presentation is confusing and difficult to follow sequentially. Key concepts are introduced without adequate explanation or forward references, forcing readers to jump ahead (or stay confused even after doing so).
  - "Generative optimization" is not a widely adopted term. It is under-explained and lacks clear references in abstract and introduction.
  - It remains unclear what exactly is being optimized (e.g., what the agent’s parameters are and how they are represented in what search space).
  - The purpose and role of "templates" are ambiguous: why three of them are introduced, whether they are used separately or jointly, and how they relate to the learning setup.
  - The term "meta-graph" is inconsistently formatted (sometimes "meta graph") and poorly defined.
- Significant handcrafting undermines generality: different workflows are manually designed for each environment, and template selection appears domain-specific with no clear decision criteria.
- Baseline choices are questionable. Comparing to traditional RL methods (e.g., DQN, PPO) in Atari might be misleading given the fundamentally different paradigms. Stronger comparisons would be other LLM-based optimization frameworks (e.g., DSPy, TextGrad, OptoPrime), which are mentioned in related work but not evaluated against.

**Questions:**

see above weakness

---

### Note · Authors · 2026-01-05

**Comment:**

We are really grateful for the comments and feedback, especially the hours of work reviewers and ACs have contributed to improving our paper. We have made significant improvements in readability of the paper and highlighted our contribution in a significantly revised version of the paper. Therefore, we are withdrawing the current version. Thank you guys very much for the hardwork and we really appreciate it!

**Withdrawal Confirmation:**

I have read and agree with the venue's withdrawal policy on behalf of myself and my co-authors.